# Recursive Entropic Risk Optimization in Discounted MDPs: Sample Complexity Bounds with a Generative Model

**Oliver Mortensen**
*Department of Computer Science*
*University of Copenhagen*

*olmo@di.ku.dk*

**Mohammad Sadegh Talebi**
*Department of Computer Science*
*University of Copenhagen*

*sadegh.talebi@di.ku.dk*

**Reviewed on OpenReview:** *https: // openreview. net/ forum? id= TFwSG4uYwl*

## Abstract

We study risk-sensitive reinforcement learning in finite discounted infinite-horizon Markov Decision Processes with recursive entropic risk measures (ERM), where the risk parameter $\beta \neq 0$ controls the agent's risk attitude: $\beta > 0$ for risk-averse behavior and $\beta < 0$ for risk-seeking behavior. A generative model of the MDP is assumed to be available. Our focus is on the sample complexities of learning the optimal state–action value function (value learning) and a near-optimal policy (policy learning) under recursive ERM. We introduce a model-based algorithm, called Model-Based ERM Q-Value Iteration (MB-ERM-QVI), and derive PAC-type bounds on its sample complexity for both value and policy learning. Both PAC bounds scale exponentially with $|\beta|/(1-\gamma)$, where $\gamma$ is the discount factor. We also establish corresponding lower bounds for both value and policy learning, showing that exponential dependence on $|\beta|/(1 - \gamma)$ is unavoidable in the worst case. The bounds are tight in the number of states and actions ($S$ and $A$), providing the first rigorous sample complexity guarantees for recursive ERM across both risk-averse and risk-seeking regimes.

## 1 Introduction

In reinforcement learning (RL), the aim of the agent is to conventionally maximize the expected return, which is defined in terms of a (possibly discounted) sum of rewards (Sutton and Barto, 1998). The environment is typically modeled via the Markov Decision Process (MDP) framework (Puterman, 2014), wherein efficient computation of an optimal policy, thanks to optimal Bellman equations, renders possible. However, as a *risk-neutral* objective, the expected return is inadequate to capture the true needs of many high-stake applications arising in, e.g., medical treatment (Ernst et al., 2006), finance (Scutella and Recchia, 2013; Bielecki and Pliska, 1999), operations research (Delage and Mannor, 2010), and transportation (Kamran et al., 2020). In these domains, decision making must take into account the variability of returns, and risks thereof. To address this limitation, one may opt to maximize a risk measure of the return distribution. Alternatively, one may model the entire distribution of return, as in distributional RL (Bellemare et al., 2023), which has received significant attention over the last decade. In this paper, we focus on the former.

Within the first approach, the risk is elegantly quantified via concave risk measures, which lead to well-defined optimization problems. Notable risk measures include mean-variance (Li and Ng, 2000), value-at-risk (VaR) (Dempster, 2002), Conditional VaR (CVaR) (Shapiro et al., 2021), entropic risk (Howard and Matheson, 1972), and entropic VaR (EVaR) (Ahmadi-Javid, 2012), all of which have been applied to a wide-range of scenarios. Among these, CVaR has become particularly popular for modeling risk-sensitivity in MDPs (Chow and Ghavamzadeh, 2014; Bisi et al., 2022; Brown et al., 2020; Bäuerle and Ott, 2011), mainly due to a delicate control it offers for the undesirable tail of return distribution. Despite its popularity and appealing

interpretation, learning in MDPs with CVaR-based objectives may pose technical challenges (Bäuerle and Ott, 2011). ERM, as another popular notion, has long been considered for risk-sensitive control in MDPs and RL (Howard and Matheson, 1972; Borkar and Meyn, 2002; Hau et al., 2023b; Hu and Leung, 2023; Fei et al., 2020). However, much of the existing literature focuses on undiscounted settings, despite the prevalence of discounted MDPs; see, e.g., (Bäuerle and Rieder, 2014; Hau et al., 2023b; Mihatsch and Neuneier, 2002) for notable exceptions.

In risk-sensitive RL with a specified risk measure, objectives can be formulated in two fundamentally different ways, depending on how the risk functional is applied to the reward sequence $(r_t)_{t \geq 0}$. The first approach, referred to as *non-recursive* (also called non-iterated or static), consists in directly applying the risk functional to the total return (e.g., $\sum_{t=0}^{\infty} \gamma^t r_t$ in the discounted case) (Borkar, 2002; Borkar and Meyn, 2002; Hau et al., 2023a). The second, termed *recursive* (also called iterated, nested, or dynamic), applies the risk functional at every step $t$ to the reward-to-go (Asienkiewicz and Jaśkiewicz, 2017; Bäuerle and Glauner, 2022; Deng et al., 2025); see Section 3 for details. In other words, the non-recursive formulation captures trajectory-level risk, whereas the recursive formulation deals with risk locally at each step. The two formulations are qualitatively different and should be viewed as orthogonal modeling choices. While trajectory-level risk is often easier to interpret, it may lead to policies that allow the agent to visit high-risk states, even though the risk of the entire trajectory is still controlled, which might be unacceptable in many safety-critical applications. In contrast, the recursive formulation may lead to more cautious behavior by discouraging entry into high-risk states at every step, which can be either desirable or overly conservative depending on the application (Deng et al., 2025). From a theoretical perspective, another key distinction is that non-recursive formulations do not generally admit Bellman-type optimality equations and may lead to time-inconsistent optimal policies (see Jaquette (1976)), whereas recursive formulations preserve Bellman-type optimality structures. Motivated by these considerations, we study risk-sensitive discounted RL with objectives defined via the recursive ERM.

## 1.1 Main Contributions and Paper Organization

We consider risk-sensitive RL in tabular discounted MDPs under recursive ERM, encompassing both the risk-averse and risk-seeking regimes. We assume that the agent has access to a generative model of the MDP, namely, a simulator that generates samples from the true MDP for arbitrary state-action pairs. Learning performance is assessed in terms of sample complexity, defined as the total number $T$ of samples required, for given $(\varepsilon, \delta)$, to obtain either an $\varepsilon$-optimal policy (*policy learning*) or an $\varepsilon$-close approximation of the optimal Q-value in the max-norm (*value learning*), with probability exceeding $1 - \delta$.

We make the following contributions. We develop a model-based algorithm, called Model-Based ERM Q-Value Iteration (MB-ERM-QVI), and establish PAC-type bounds on its sample complexity for both value learning and policy learning. MB-ERM-QVI is based on a plug-in estimation of the transition kernel combined with a Q-value iteration scheme adapted to recursive ERM objectives. This QVI structure is inspired by the value iteration method of (Bäuerle and Glauner, 2022), which considers the risk-averse planning setting with known dynamics. We show that the sample complexity of MB-ERM-QVI for value learning (Theorem 1) and policy learning (Theorem 2) scales as[1]

$$\widetilde{O}\left(\frac{SA}{\varepsilon^2(1-\gamma)^2|\beta|^2}e^{2|\beta|/(1-\gamma)}\right) \quad \text{and} \quad \widetilde{O}\left(\frac{SA}{\varepsilon^2(1-\gamma)^2|\beta|^2}\min\left\{S, \frac{1}{(1-\gamma)^2}\right\}e^{2|\beta|/(1-\gamma)}\right),$$

respectively. These bounds hold for any discounted MDP with $S$ states, $A$ actions, discount factor $\gamma$, and risk parameter $\beta$, with $\beta > 0$ (respectively, $\beta < 0$) corresponding to a risk-averse (respectively, risk-seeking) agent; see Section 3 for a precise definition. Moreover, the bounds are valid over the entire $\varepsilon$-range, namely $\varepsilon \in (0, \frac{1}{1-\gamma}]$. A notable feature of these results is the exponential dependence on the effective horizon $1/(1-\gamma)$, which is absent in the conventional risk-neutral setting, wherein $\beta = 0$.

---

[1]Throughout the paper, $\widetilde{\mathcal{O}}$ and $\widetilde{\Omega}$ suppress logarithmic factors in the relevant problem parameters.

Table 1: Summary of upper and lower bounds presented in this paper. $\beta$ denotes the risk parameter, where $\beta > 0$ (respectively, $\beta < 0$) corresponds to a risk-averse (respectively, risk-seeking) agent. All ERM bounds are from this paper. See (Gheshlaghi Azar et al., 2013; Agarwal et al., 2020; Li et al., 2020; Jin et al., 2024) for the risk-neutral bounds.

| Problem | Upper Bound | Lower Bound |
|---|---|---|
| ERM (value learning) | $\widetilde{\mathcal{O}}\left(\frac{SA}{\varepsilon^2(1-\gamma)^2\|\beta\|^2}e^{2\|\beta\|/(1-\gamma)}\right)$ [Thm. 1] | $\widetilde{\Omega}\left(\frac{SA}{\varepsilon^2\|\beta\|^2}e^{\|\beta\|/(1-\gamma)}\right)$ [Thm. 3] |
| ERM (policy learning) | $\widetilde{\mathcal{O}}\left(\frac{SA}{\varepsilon^2(1-\gamma)^2\|\beta\|^2}\min\left\{S,\frac{1}{(1-\gamma)^2}\right\}e^{2\|\beta\|/(1-\gamma)}\right)$ [Thm. 2] | $\widetilde{\Omega}\left(\frac{SA}{\varepsilon^2\|\beta\|^2}e^{\|\beta\|/(1-\gamma)}\right)$ [Thm. 4] |
| Risk-neutral (value learning) | $\widetilde{\mathcal{O}}\left(\frac{SA}{\varepsilon^2(1-\gamma)^3}\right)$ | $\widetilde{\Omega}\left(\frac{SA}{\varepsilon^2(1-\gamma)^3}\right)$ |
| Risk-neutral (policy learning) | $\widetilde{\mathcal{O}}\left(\frac{SA}{\varepsilon^2(1-\gamma)^3}\right)$ | $\widetilde{\Omega}\left(\frac{SA}{\varepsilon^2(1-\gamma)^3}\right)$ |

We further establish worst-case lower bounds on the sample complexity of recursive ERM. Specifically, we show that for value learning (Theorem 3) and policy learning (Theorem 4), at least

$$\widetilde{\Omega}\left(\frac{SA}{\varepsilon^2\|\beta\|^2}e^{\|\beta\|/(1-\gamma)}\right)$$

samples are required to achieve $\varepsilon$-optimality. These lower bounds demonstrate that the exponential dependence on $\|\beta\|/(1-\gamma)$ in sample complexity upper bounds is unavoidable in the worst case, thereby establishing that learning under recursive ERM is fundamentally more challenging than in the risk-neutral case. To the best of our knowledge, these results constitute the first upper and lower bounds on the sample complexity of recursive ERM in discounted MDPs. A summary of our results is provided in Table 1.

The remainder of this paper is organized as follows. Section 2 reviews related work. Section 3 introduces the necessary background and formal problem setup. Section 4 presents the MB-ERM-QVI algorithm, while Section 5 reports its sample complexity guarantees, with proofs deferred to Section 6. Lower bounds are presented in Section 7. Section 8 presents numerical results to demonstrate the performance of MB-ERM-QVI. Finally, Section 9 concludes with a discussion and directions for future research. Additional background on risk measures, along with omitted proofs, is provided in the appendix.

## 2 Related Work

**Risk-neutral discounted RL.** There is a large body of papers on provably-sample efficient learning algorithms in tabular discounted MDPs, encompassing a variety of settings such as the generative setting (Kakade, 2003; Gheshlaghi Azar et al., 2013; Agarwal et al., 2020; Sidford et al., 2018; Li et al., 2020; Jin et al., 2024), the offline (or batch) setting (Rashidinejad et al., 2021; Li et al., 2024a), and the online setting (Strehl and Littman, 2008; Lattimore and Hutter, 2014). In our overview of risk-neutral work, we restrict attention to the generative setting —which is the setting we consider— with the aim of collecting most notable developments and key results. In this line, Kearns and Singh (1998) report the earliest known sample complexity bounds, which is achieved by a model-free method. Gheshlaghi Azar et al. (2013) substantially improve upon this by showing that a simple model-based method attains optimal sample complexity bounds scaling as $\widetilde{\mathcal{O}}\left(\frac{SA}{\varepsilon^2(1-\gamma)^3}\right)$ for both value learning and policy learning, albeit for substantially limited $\varepsilon$-ranges. It also establishes a lower bound of $\widetilde{\Omega}\left(\frac{SA}{\varepsilon^2(1-\gamma)^3}\right)$ for value learning. Further algorithms and results are reported in more recent subsequent work, notably including (Sidford et al., 2018; Wang, 2020; Li et al., 2024b; Jin et al., 2024). Among these, Sidford et al. (2018); Wang (2020); Jin et al. (2024) present model-free methods, with Sidford et al. (2018) and Jin et al. (2024) presenting minimax-optimal bounds, although valid for restricted $\varepsilon$-ranges. Under model-based methods, minimax-optimal bounds, beyond (Gheshlaghi Azar et al., 2013), are reported by Agarwal et al. (2020); Li et al. (2024b). In particular, Agarwal et al. (2020) use empirical MDP combined with a black-box planner, and report a minimax-optimal bound for $\varepsilon \in \left(0, \frac{1}{\sqrt{1-\gamma}}\right)$, thus expanding that in (Gheshlaghi Azar et al., 2013). More recently, Li et al. (2024b) establish minimax-optimal bounds for the entire $\varepsilon$-range, which are achieved by model-based methods built via the empirical MDP but

with reward perturbations or conservative planning. It is worth emphasizing that existing optimal sample complexities rely on techniques that crucially exploit the additivity of the return with respect to rewards; this structural property generally fails for risk-sensitive objectives, and the corresponding techniques do not carry over.

We remark that the abovementioned works look at the learning performance in a worst-case scenario, which yield sample complexity bounds that hold for a model class. This is typically done via uniformly sampling various state-action pairs. In contrast, some studies (e.g., Al Marjani and Proutiere (2021); Zanette et al. (2019)) consider adaptive sampling to account for the heterogeneity across state-action space of the MDP, typically resulting in instance-dependent bounds.

**Risk-sensitive RL.** There exists a substantial literature on decision making under a risk measure in bandit and RL settings. In bandits, risk-sensitive objectives are typically studied through regret minimization; see, e.g., (Sani et al., 2012; Maillard, 2013; Khajonchotpanya et al., 2021). Extensions to MDPs introduce substantially richer structural and algorithmic challenges, which are the focus of this work. The literature on RL under risk measures may be broadly categorized by the type of risk measure studied. Representative examples include CVaR (Deng et al., 2025; Du et al., 2023; Chen et al., 2024; Lam et al., 2022), ERM (Borkar and Meyn, 2002; Moharrami et al., 2025; Marthe et al., 2023; 2025; Hau et al., 2023b), mean-variance risk (Sood et al., 2023; Huang et al., 2022; La and Ghavamzadeh, 2013), and EVaR (Ni and Lai, 2022; Ganguly et al.). Among these, CVaR has been the most extensively studied. Under non-recursive CVaR, Du et al. (2023) and Chen et al. (2024) investigate online episodic RL in the regret setting for tabular MDPs and MDPs with function approximation, respectively, while reward-free RL is studied by Ni et al. (2024). Under recursive CVaR, Deng et al. (2025) analyze sample complexity in the generative setting; however, their analysis relies on structural properties specific to CVaR and does not extend to ERM.

ERM has also been widely studied, beginning with early work such as (Howard and Matheson, 1972) and followed by a rich literature across diverse settings (Borkar and Meyn, 2002; Borkar, 2001; 2002; Sadana et al., 2024; Fei et al., 2020; 2021b;a; Marthe et al., 2023; 2025; Moharrami et al., 2025; Hau et al., 2023b). Under non-recursive ERM, most work focuses on the average-reward or episodic settings, with planning studied in (Howard and Matheson, 1972; Borkar and Meyn, 2002; Marthe et al., 2025) and learning in (Borkar, 2002; Moharrami et al., 2025; Marthe et al., 2023). The discounted setting has received comparatively little attention, largely due to technical challenges introduced by discounting; notable exceptions include planning results by Bäuerle and Rieder (2014) and Hau et al. (2023b) and learning results by Mihatsch and Neuneier (2002), which modify the problem formulation to address time inconsistency. Under recursive ERM, recent works such as (Fei et al., 2020; 2021a;b; Hu and Leung, 2023; Liang and Luo, 2024) study online episodic RL in the regret setting. To the best of our knowledge, existing work on discounted MDPs under recursive ERM is limited to planning; a notable example is (Bäuerle and Glauner, 2022), which provides a thorough theoretical treatment but does not propose learning algorithms. The analysis of Bäuerle and Glauner (2022) (and the works in the known-model setting) is limited to the case of known models, where the problem does not involve statistical estimation. As a result, sample complexity analyses (under policy or value learning), which aim to characterize statistical hardness, are not relevant in their setting.

Some work in risk-sensitive RL and control studies broader classes of risk measures. Two notable classes studied in this context are coherent risk measures and optimized certainty equivalent (OCE) measures, both of which include important special cases such as mean–variance and CVaR. While ERM is not coherent, it belongs to the OCE class for $\beta > 0$; a brief overview of risk measures is provided in Appendix A. Results for coherent risks (Petrik and Subramanian, 2012; Tamar et al., 2015; Lam et al., 2022; Zhao et al., 2024) do not apply to ERM, and existing results for OCE risks (Wang et al., 2025; Xu et al., 2023; Rigter et al., 2023) do not address provably sample-efficient learning under recursive ERM in discounted MDPs. In particular, Rigter et al. (2023) consider offline RL in discounted MDPs under recursive OCE but do not provide sample-complexity guarantees. We also note that a connection between MDPs with recursive coherent risks and distributionally robust MDPs has been established by Bäuerle and Glauner (2022). Finally, we note that approaches such as safe RL and constrained MDPs (Chow et al., 2018; Chow and Pavone, 2014) incorporate risk awareness into policies via constraints, but without altering the definition of return; they are therefore generally regarded as orthogonal to the present setting.

# 3 Background

**Notations.** For $n \in \mathbb{N}$, let $[n] := \{1, \ldots, n\}$. $\mathbb{1}_A$ denotes the indicator function of an event $A$. Given a set $\mathcal{X}$, $\Delta(\mathcal{X})$ denotes the probability simplex over $\mathcal{X}$. We use the convention that $\|\cdot\| := \|\cdot\|_\infty$ and explicitly use the subscript $\|\cdot\|_p$ when using $p$-norms for $1 \le p < \infty$. The notation $L^\infty(\Omega, \mathcal{F}, \mathbb{P})$ denotes the space of essentially bounded random variables on the probability space $(\Omega, \mathcal{F}, \mathbb{P})$.

## 3.1 Entropic Risk Preferences

The entropic risk measure (ERM) is rooted in expected utility theory (Mas-Colell et al., 1995). Consider for $\beta \ne 0$ the class of utility functions $u(t) = \frac{1}{\beta}(1 - e^{-\beta t})$ defined for $t \in \mathbb{R}$. The utility $u$ is supposed to describe the preferences of some economic agent in the form of how much utility $u(t)$ she derives from some monetary quantity $t \in \mathbb{R}$. For any bounded random variable $X \in L^\infty(\Omega, \mathcal{F}, \mathbb{P})$, it is easy to verify that the associated *certainty equivalent* to $u$ is $u^{-1}(\mathbb{E}[u(X)]) = \frac{-1}{\beta} \log(\mathbb{E}[e^{-\beta X}])$, which expresses the amount of money that would give the same utility as that of entering in the bet given by the random variable $X$. We thus define the functional $\rho : L^\infty(\Omega, \mathcal{F}, \mathbb{P}) \to \mathbb{R}$ by[2]

$$\rho(X) := \rho(X; \beta) := -\frac{1}{\beta} \log\left(\mathbb{E}[e^{-\beta X}]\right). \tag{1}$$

Evidently, when $\beta \to 0$ we recover the risk-neutral case, which simply coincides with the expectation: $\lim_{\beta \to 0} \rho(X) = \mathbb{E}[X]$. Further, it is straightforward to see that $\rho$ admits the following:

$$\rho(X) \le \rho(Y), \quad \text{for any } X \le Y, \tag{2}$$

$$\rho(c) = c, \quad \text{for any } c \in \mathbb{R}, \tag{3}$$

$$\rho(X) \le \mathbb{E}[X], \quad \text{for } \beta > 0, \tag{4}$$

$$\rho(X) \ge \mathbb{E}[X], \quad \text{for } \beta < 0, \tag{5}$$

where properties (4)-(5) follow from Jensen's inequality. Using $\rho$ as a measure of the preference for different random variables, it follows directly from (3)-(5) that $\rho(X) \le \rho(\mathbb{E}[X])$ for $\beta > 0$ and that $\rho(X) \ge \rho(\mathbb{E}[X])$ for $\beta < 0$. It further shows that $\beta > 0$ is associated with *risk-aversion*, while $\beta < 0$ is associated with a *risk-seeking* behavior. It is well-known that ERM, unlike CVaR, does not belong to the nice class of coherent risk measures; we refer the reader to Appendix A for a primer on risk measures, where we collect some definitions and concrete examples.

## 3.2 Discounted Markov Decision Processes with Entropic Risk

We write the 6-tuple $M = (\mathcal{S}, \mathcal{A}, P, R, \gamma, \beta)$ to denote a finite, discounted infinite-horizon MDP, where $\mathcal{S} = \{1, 2, \ldots, S\}$ is the finite state space of size $S := |\mathcal{S}|$, $\mathcal{A} = \{1, 2, \ldots, A\}$ is the finite action space of size $A := |\mathcal{A}|$, $P : \mathcal{S} \times \mathcal{A} \to \Delta(\mathcal{S})$ is the transition probability function, $R : \mathcal{S} \times \mathcal{A} \to [0, 1]$ is the reward function, $\gamma \in (0, 1)$ is the discount factor, and $\beta \ne 0$ is the risk-parameter. We use $Z = \mathcal{S} \times \mathcal{A}$ to denote the set of all state-action pairs, and write $P_{s,a}(s')$ as short-hand notation for $P(s'|s, a)$ for any $(s, a) \in Z$. For simplicity of exposition, we consider a deterministic reward function, as is standard in the literature. The agent interacts with the MDP $M$ as follows. At the beginning of the process, $M$ is in some initial state $s_0 \in \mathcal{S}$. At each time $t \ge 0$, the agent is in state $s_t \in \mathcal{S}$ and decides on an action $a_t \in \mathcal{A}$ according to some rule. The MDP generates a reward $r_t := R(s_t, a_t)$ and a next-state $s_{t+1} \sim P(\cdot|s_t, a_t)$. The MDP moves to $s_{t+1}$ when the next time slot begins, and this process continues ad infinitum. This process yields a growing sequence $(s_t, a_t, r_t)_{t \ge 0}$. The agent's goal is to maximize an objective function, as a function of the collected rewards $(r_t)_{t \ge 0}$, which depends on both $\gamma$ and $\beta$.

To concretely define the agent's objective using ERM, we discuss two approaches of applying the functional $\rho$ in (1) in the context of MDPs. The first approach, called *non-recursive* (or static or non-iterated) (Hau et al.,

---

[2]We note that there is a lack of consensus regarding the sign of $\beta$ in the definition of ERM. We follow this convention considering its widespread use in the actuarial literature (Asienkiewicz and Jaśkiewicz, 2017). We refer to Appendix A for a related discussion.

2023b; Marthe et al., 2025), consists in applying $\rho$ to the total discounted sum of rewards $\rho\left(\sum_{t=0}^{\infty}\gamma^{t}r_{t}\right)$, which is well-defined under the bounded rewards assumption, i.e., $r_t \in [0,1]$ for all $t$. This problem admits no obvious optimality equation, although solution and approximation schemes for planning have been proposed in the literature (Hau et al., 2023b; Marthe et al., 2025). The other approach, wherein $\rho$ is applied at every step, is called *recursive* (also called dynamic or iterated) (Asienkiewicz and Jaśkiewicz, 2017). The planning problem in this case is tractable thanks to existence of Bellman-type optimality equations. The recursive approach also guarantees the existence of an optimal stationary deterministic policy, whereas the non-recursive approach may lead to optimal policies that are not time-consistent (see Jaquette (1976)). In this paper, we study the case where the RL objective is defined via the recursive ERM.

### 3.3 Value Function and Q-function

We shall introduce some notations and definitions to formally define the value function $V$ and state-action value function $Q$ (henceforth, $Q$-value) of a policy. We follow the approach of Asienkiewicz and Jaśkiewicz (2017) and Bäuerle and Jaśkiewicz (2024), but since none of their cases include our $\beta < 0$ case and also only cover value function, we present in Appendix B the full setup with history-dependent policies as well as a thorough definition of the value and $Q$-values. There, we prove existence of a stationary optimal policy, and show that the value functions of this policy satisfy a Bellman optimality equation and that the value of any policy satisfies a Bellman recursion. We give an outline here that only deals with stationary policies, which is justified by the results of Appendix B.

Let $v \in \mathbb{R}^S$ and $\pi : \mathcal{S} \to \mathcal{A}$ be a stationary deterministic policy. We define $\rho_{s,a} : \mathbb{R}^S \to \mathbb{R}$ as

$$\rho_{s,a}(v) = -\frac{1}{\beta} \log\left( \mathbb{E}_{s' \sim P_{s,a}}[e^{-\beta v(s')}] \right) \tag{6}$$

and slightly abusing the notation, we write $\rho_{s,\pi}$ when $a = \pi(s)$, i.e., $\rho_{s,\pi} := \rho_{s,\pi(s)}$. We then introduce the operator $J_\pi : \mathbb{R}^S \to \mathbb{R}^S$ with $J_\pi(v)(s) = R(s,\pi(s)) + \gamma\rho_{s,\pi}(v)$. The $N$-step total discounted utility $J_N(s,\pi)$ is defined by applying $J_\pi$ recursively $N$ times to the 0-map: $J_N(s,\pi) := J_\pi^N(\mathbf{0})(s)$. Note that the outer-most iteration corresponds to the immediate time-step. Then, the value of policy $\pi$ is defined as: $V^\pi(s) = \lim_{N \to \infty} J_N(s,\pi)$ for all $s \in \mathcal{S}$. By properties (2)-(3), it follows that $J_N(s,\pi)$ is increasing in $N$ and that $J_N(s,\pi) \leq \frac{1}{1-\gamma}$ for all $s \in \mathcal{S}$, so that the limit above exists and the value function is thus well-defined. The optimal state-values are defined as $V^*(s) = \sup_\pi V^\pi(s)$ for all $s \in \mathcal{S}$, where the sup is taken over all possible policies. Any policy achieving $V^*(s)$ for all $s \in \mathcal{S}$ is called optimal, and $V^*$ is called the optimal value function. Further, given $\varepsilon > 0$, a policy achieving $V^\pi(s) \geq V^*(s) - \varepsilon$ for all $s \in \mathcal{S}$ is called $\varepsilon$-optimal.

Asienkiewicz and Jaśkiewicz (2017) consider a more general MDP framework that is not restricted to finite MDPs or stationary policies; they prove that under some conditions —that are trivially fulfilled in the case of finite MDPs— there exists a stationary deterministic optimal policy in the $\beta > 0$ case. This result is easily extended to the $\beta < 0$ case, and we give a unified proof for completeness in Appendix B. The optimal value function $V^*$ satisfies the Bellman optimality equation:

$$V^*(s) = \max_{a \in \mathcal{A}} \left( R(s,a) - \frac{\gamma}{\beta} \log\left( \mathbb{E}_{s' \sim P_{s,a}}[e^{-\beta V^*(s')}] \right) \right), \quad \forall s \in \mathcal{S}.$$

Further, for any stationary deterministic policy $\pi$, the value function satisfies the Bellman recursion:

$$V^\pi(s) = R(s,\pi(s)) - \frac{\gamma}{\beta} \log\left( \mathbb{E}_{s' \sim P_{s,\pi(s)}}[e^{-\beta V^\pi(s')}] \right), \quad \forall s \in \mathcal{S}. \tag{7}$$

We introduce the $Q$-value functions using a similar approach. Given $\pi : \mathcal{S} \to \mathcal{A}$, we define the operator $L_\pi : \mathbb{R}^S \to \mathbb{R}^{S \times A}$ as follows: for all $v : \mathcal{S} \to \mathbb{R}$, for all $(s,a)$, $L_\pi(v)(s,a) = R(s,\pi(s)) + \gamma\rho_{s,\pi}(v)$. Also, we define the operator $L : \mathbb{R}^S \to \mathbb{R}^{S \times A}$ as follows: for all $v : \mathcal{S} \to \mathbb{R}$, for all $(s,a)$, $L(v)(s,a) = R(s,a) + \gamma\rho_{s,a}(v)$. We define the $N$-step total discounted utility of the state-action pair $(s,a)$ under $\pi$ as $L_N(s,a,\pi) := (L \circ J_\pi^{N-1}(\mathbf{0}))(s,a)$, and the limit is denoted $Q^\pi(s,a)$: $Q^\pi(s,a) = \lim_{N \to \infty} L_N(s,a,\pi)$. Although Asienkiewicz and Jaśkiewicz (2017) do not consider $Q$-value functions, building on their arguments we show in Appendix B that it suffices

to consider stationary policies when solving $\max_\pi Q^\pi(s,a) =: Q^*(s,a)$ for all $(s,a)$ and that $Q^*$ satisfies the optimality equation:

$$Q^*(s,a) = R(s,a) - \frac{\gamma}{\beta} \log\left(\mathbb{E}_{s' \sim P_{s,a}}\left[e^{-\beta \max_{a'} Q^*(s',a')}\right]\right), \quad \forall(s,a) \in \mathcal{S} \times \mathcal{A}.$$

Similarly, the $Q$-value of a policy $\pi$ satisfies the Bellman recursion:

$$Q^\pi(s,a) = R(s,a) - \frac{\gamma}{\beta} \log\left(\mathbb{E}_{s' \sim P_{s,a}}\left[e^{-\beta Q^\pi(s',\pi(s'))}\right]\right), \quad \forall(s,a) \in \mathcal{S} \times \mathcal{A}.$$

Let us introduce the operators $\mathcal{T}^\pi, \mathcal{T} : \mathbb{R}^{S \times A} \to \mathbb{R}^{S \times A}$, which for $f : \mathcal{S} \times \mathcal{A} \to \mathbb{R}$ are defined as

$$(\mathcal{T}f)(s,a) = R(s,a) - \frac{\gamma}{\beta} \log \sum_{s'} P_{s,a}(s') e^{-\beta \max_{a'} f(s',a')},$$

$$(\mathcal{T}^\pi f)(s,a) = R(s,a) - \frac{\gamma}{\beta} \log \sum_{s'} P_{s,a}(s') e^{-\beta f(s',\pi(s'))}.$$

Then, the Bellman equations above can be written as $Q^* = \mathcal{T}Q^*$ and $Q^\pi = \mathcal{T}^\pi Q^\pi$.[3]

We conclude this section by a remark about the case of rewards bounded in $[R_{\min}, R_{\max}]$.

**Remark 1.** *For rewards bounded in $[R_{\min}, R_{\max}]$, one can equivalently consider rewards in $[0,1]$ but with a risk parameter $\frac{\beta}{w}$, with $w := R_{\max} - R_{\min}$. This is verified by observing that (i) $\rho$ is translation invariant, implying that the range $w$ is important –not the absolute values– so that one must model rewards as belonging to $[0,w]$; and (ii) one has $\rho(wX; \beta) = -\frac{w}{w\beta} \log(\mathbb{E}[e^{-\beta wX}] = w\rho(Xw; \beta)$ for any $w > 0$, so that scaling the range by $w$ amounts to working with $\rho$ with a risk parameter $w\beta$. This observation justifies our focus on rewards in $[0,1]$, but more importantly signals that the risk parameter is calibrated to a specific reward range.*

### 3.4 Learning Performance

We consider RL algorithms that aim to find an $\varepsilon$-optimal policy or an $\varepsilon$-optimal value function for input $\varepsilon > 0$ under ERM, while having access to a generative model (or simulator) of the MDP. Precisely speaking, the generative model can produce a sample $s' \sim P_{s,a}$ for any queried state-action $(s,a)$. We consider two types of such algorithms, which we generically denote by $\mathcal{U}$: The first type outputs a $Q$-value $Q_T^{\mathcal{U}} : \mathcal{S} \times \mathcal{A} \to \mathbb{R}$, whereas the second outputs a policy $\pi_T^{\mathcal{U}} : \mathcal{S} \to \mathcal{A}$ using $T$ samples. We evaluate the quality of an algorithm that outputs a $Q$-value by $\|Q^* - Q_T^{\mathcal{U}}\|$. For an algorithm that instead outputs a policy, we evaluate it in terms of $\|V^* - V^{\pi_T^{\mathcal{U}}}\|$. Often, we will suppress $T$ from the notation. This leads to the notion of $(\varepsilon, \delta)$-correct value and policy for input parameters $(\varepsilon, \delta)$, which we borrow from the literature on risk-neutral RL:

**Definition 1** ($(\varepsilon, \delta)$-correct value and policy). *An algorithm $\mathcal{U}$ that outputs a $Q$-value $Q^{\mathcal{U}}$ is called $(\varepsilon, \delta)$-value-correct on a set of MDPs $\mathbb{M}$ if $\mathbb{P}(\|Q^* - Q^{\mathcal{U}}\| \leq \varepsilon) \geq 1 - \delta$ for all $M \in \mathbb{M}$. Similarly, an algorithm $\mathcal{U}$ that outputs a policy $\pi^{\mathcal{U}}$ is called $(\varepsilon, \delta)$-policy-correct on a set of MDPs $\mathbb{M}$ if $\mathbb{P}(\|V^* - V^{\pi^{\mathcal{U}}}\| \leq \varepsilon) \geq 1 - \delta$ for all $M \in \mathbb{M}$.*

The notion of $(\varepsilon, \delta)$-value-correctness yields a sample complexity notion in the case of *value learning*, while $(\varepsilon, \delta)$-policy-correctness serves a similar role for *policy learning*.

We remark that any algorithm that outputs a $Q$-value also outputs a policy, e.g., the greedy policy with respect to the output $Q$-value. However, it is well-known that such a greedy policy can be off by a factor of $1/(1 - \gamma)$, which impacts the corresponding sample complexity of policy learning; see discussions in (Singh and Yee, 1994; Sidford et al., 2018; Agarwal et al., 2020). To remedy this, the literature resort to proof arguments, which may however come at a cost of limiting the $\varepsilon$-range, as briefly discussed in Section 2.

---

[3]We note that our analysis only rests on the Bellman optimality equation; the Bellman equations for $V^\pi$ and $Q^\pi$ are included for completeness.

## 4 Model-Based ERM Q-Value Iteration

In this section, we present a model-based algorithm, called MB-ERM-QVI, for value and policy learning settings under ERM, assuming access to a generative model of the MDP. Then, we derive PAC-type bounds on its sample complexity.

We begin with introducing the protocol for obtaining $N$ samples from each state-action pair in $Z = \mathcal{S} \times \mathcal{A}$; this is done by making a total of $T = NSA$ calls to the generative model (see Algorithm 1).

---

**Algorithm 1:** Model estimation

**Input:** Generative model $P$
**Output:** Model estimate $\widehat{P}$
1 **Function** EstimateModel($N$):
2     $\forall (s,z) \in \mathcal{S} \times Z : m(s,z) = 0$
3     **for** *each $z \in Z$* **do**
4         **for** $i = 1, 2, \ldots, N$ **do**
5             $s \sim P(\cdot|z)$
6             $m(s,z) := m(s,z) + 1$
7         **end**
8         $\forall s \in \mathcal{S} : \widehat{P}(s,z) = \frac{m(s,z)}{N}$
9     **end**
10     **return** $\widehat{P}$

---

**Algorithm 2:** ERM-QVI (for empirical MDP)

**Input:** Empirical MDP
1 $\widehat{M} = (\mathcal{S}, \mathcal{A}, \widehat{P}, R, \gamma, \beta)$, number $k$ of iterations
**Output:** Estimate $Q_k$ of optimal $Q$-function $Q^*$
2 Initialization: $\forall (s,a)$ set $Q(s,a) = 0$
3 **for** $j = 0, 1, \ldots, k-1$ **do**
4     **for** *all $(s,a) \in \mathcal{S} \times \mathcal{A}$* **do**
5         $Q_{j+1}(s,a) =$
            $R(s,a) - \frac{\gamma}{\beta} \log \left( \mathbb{E}_{s' \sim \widehat{P}_{s,a}} \left[ e^{-\beta \max_{a'} Q_j(s',a')} \right] \right)$
6     **end**
7 **end**
8 $\forall s \in \mathcal{S} : \pi_k(s) = \operatorname{argmax}_{a \in \mathcal{A}} Q_k(s,a)$
9 **return** $Q_k$ and $\pi_k$

---

Let $\widehat{P}$ denote the plug-in estimator built using the $T = NSA$ independent samples obtained from the generative model; that is, for $(s,a,s') \in Z \times \mathcal{S}$, $\widehat{P}(s'|s,a) = \frac{n(s,a,s')}{N}$, where $n(s,a,s')$ denotes the number of times $s'$ was observed under the queried pair $(s,a) \in Z$. The model-based approach we describe relies on the empirical MDP formulated using $\widehat{P}$, $\widehat{M} = (\mathcal{S}, \mathcal{A}, R, \widehat{P}, \gamma, \beta)$, but it is otherwise general in the sense that it can use any oracle that outputs an $\varepsilon$-optimal policy for any $\varepsilon > 0$. We prove the existence of one such oracle in the analysis (cf. Lemma 2) in the form of a $Q$-value iteration method akin to that of the classical risk-neutral setting. It is the basis for ERM-QVI (Algorithm 2), which is an empirical $Q$-value iteration for discounted MDPs with ERM with correctness guarantees.

Equipped with these, we introduce MB-ERM-QVI. For an input $\varepsilon > 0$, the algorithm consists in:

(i) building an empirical MDP $\widehat{M} = (\mathcal{S}, \mathcal{A}, \widehat{P}, R, \gamma, \beta)$ via calling the generative model $N$ times (namely, $\widehat{P} = \texttt{EstimateModel}(N)$);

(ii) running ERM-QVI with $\widehat{M}$ as input.

We show in Lemma 2 how to pick $k := k(\varepsilon)$ to ensure $\varepsilon$-value-correctness (i.e., $\|Q^* - Q_k\| \leq \varepsilon$) and $\varepsilon$-policy-correctness (i.e., $\|V^* - V^{\pi_k}\| \leq \varepsilon$).

## 5 Sample Complexity Analysis of MB-ERM-QVI

In this section, we present a sample complexity analysis of MB-ERM-QVI under both policy learning and value learning.

### 5.1 Properties of ERM-QVI

We first state two results for ERM-QVI, establishing its convergence properties. Their proofs are presented in Appendix D.

**Lemma 1** (Contraction properties). *The operators $\mathcal{T}$ and $\mathcal{T}^\pi$ are $\gamma$-contractions with respect to the max-norm, i.e., $\|\mathcal{T}f_1 - \mathcal{T}f_2\| \leq \gamma\|f_1 - f_2\|$ and $\|\mathcal{T}^\pi f_1 - \mathcal{T}^\pi f_2\| \leq \gamma\|f_1 - f_2\|$ for value functions $f_1$ and $f_2$.*

The proof of this result is very similar to that of Part (a) in Theorem 3.1 in (Asienkiewicz and Jaśkiewicz, 2017); nevertheless, we include it for completeness. The next lemma shows that for large enough $k$ in ERM-QVI, we can obtain $Q_k$ and $V^{\pi_k}$ that are as close to, respectively, $Q^*$ and $V^*$ as desired:

**Lemma 2.** *Fix $\varepsilon > 0$. Under ERM-QVI (Algorithm 2), one has: (i) $\|Q_k - Q^*\| < \varepsilon$ if $k > \frac{-\log((1-\gamma)\varepsilon)}{\log(1/\gamma)}$; and (ii) $\|V^{\pi_k} - V^*\| < \varepsilon$ if $k > \frac{\log 2 - \log((1-\gamma)^2\varepsilon)}{\log(1/\gamma)}$.*

## 5.2 Sample Complexity Upper Bounds

We are ready to present sample complexity bounds for MB-ERM-QVI: Theorem 1 states such a result for the case of value learning, while Theorem 2 offers a bound for policy learning.

**Theorem 1** (Sample complexity, value learning). *For any $\varepsilon > 0$, $\delta \in (0,1)$, and any MDP $M$ with $S$ states and $A$ actions, if the learner makes*

$$T \geq \frac{2SA\gamma^2}{\varepsilon^2(1-\gamma)^2}\left(\frac{e^{|\beta|/(1-\gamma)}-1}{|\beta|}\right)^2 \log\left(\frac{SA}{\delta}\right)$$

*model calls to the generative model, then $\mathbb{P}(\|Q^* - Q_k\| \leq \varepsilon) \geq 1 - \delta$.*

**Theorem 2** (Sample complexity, policy learning). *For any $\varepsilon > 0$, $\delta \in (0,1)$, and any MDP $M$ with $S$ states and $A$ actions, if the learner makes*

$$T \geq \frac{9SA\gamma^2}{\varepsilon^2(1-\gamma)^2}\left(\frac{e^{|\beta|/(1-\gamma)}-1}{|\beta|}\right)^2 \min\left\{\frac{\gamma^2}{(1-\gamma)^2}\log\left(\frac{4SA}{\delta}\right), \log\left(\frac{4SA|\Pi|}{\delta}\right)\right\}$$

*model calls, then $\mathbb{P}(\|V^* - V^{\pi_k}\| \leq \varepsilon) \geq 1 - \delta$. Here, $\Pi$ denotes the set of stationary deterministic policies.*

The sample complexity bound offered by Theorem 2 can be further simplified using the worst-case bound $|\Pi| \leq A^S$ to $\tilde{\mathcal{O}}\left(\frac{SA}{\varepsilon^2(1-\gamma)^2}\min\left\{S, \frac{1}{(1-\gamma)^2}\right\}L^2\right)$ with $L = \frac{1}{|\beta|}\left(e^{|\beta|/(1-\gamma)}-1\right)$. To be more precise, by including log-terms, depending on whether $S \ll 1/(1-\gamma)^2$ in the problem at hand, one may obtain a bound of

$$\mathcal{O}\left(\frac{SA}{\varepsilon^2(1-\gamma)^4}\log\left(\frac{SA}{\delta}\right)L^2\right) \quad \text{or} \quad \mathcal{O}\left(\frac{SA}{\varepsilon^2(1-\gamma)^2}\left(S + \log\left(\frac{SA}{\delta}\right)\right)L^2\right).$$

Let us remark however that in problems where $|\Pi|$ grows polynomially with $S$, one will get a substantially better bound.

**Remark 2.** *Taking the limit in the PAC bound of Theorems 1–2, as $\beta \to 0$, yields corresponding sample complexity bounds for the risk-neutral case. The resulting bound for value learning is off the optimal bound by a factor of $(1-\gamma)^{-1}$, and for policy learning by a factor of $(1-\gamma)^{-1}\min\{S, (1-\gamma)^{-2}\}$; see Table 1. It is worth stressing, however, that these implied bounds are valid for the entire $\varepsilon$-range, unlike the results in, e.g., (Gheshlaghi Azar et al., 2013; Agarwal et al., 2020; Sidford et al., 2018).*

**Remark 3.** *Existing derivations of minimax sample complexity bounds in the risk-neutral setting (e.g., (Gheshlaghi Azar et al., 2013; Agarwal et al., 2020; Sidford et al., 2018; Li et al., 2020)) rely on techniques that crucially exploit the additive structure of the return with respect to rewards, such as variance lemmas establishing Bellman consistency of the variance of cumulative discounted rewards. These tools do not extend to ERM due to its intrinsic non-linearity, and are therefore not applicable in our setting.*

## 6 Proofs: Sample Complexity Upper Bounds

In this section, we prove Theorems 1 and 2. As preliminarily, we present some results that will be used in the proofs. The first one concerns basic decompositions of the error terms associated to $V^{\pi_k}$ and $Q_k$. Let $\widehat{V}^*$ and $\widehat{Q}^*$ denote the optimal value and Q-value in $\widehat{M}$, respectively, and $\pi^*$ denote an optimal policy in $M$. Then, for any $(s,a) \in \mathcal{S} \times \mathcal{A}$,

$$Q_k(s,a) \geq Q^*(s,a) - \|\widehat{Q}^{\pi^*} - Q^*\| - \|Q_k - \widehat{Q}^*\|, \tag{8}$$

$$V^{\pi_k}(s) \geq V^*(s) - \|V^{\pi_k} - \widehat{V}^{\pi_k}\| - \|\widehat{V}^{\pi^*} - V^*\| - \|\widehat{V}^{\pi_k} - \widehat{V}^*\|. \tag{9}$$

These follow from standard techniques, but for completeness, we derive them in Lemma 6 in Appendix C. In (8), the term $\|\widehat{Q}^{\pi^*} - Q^*\|$ captures the statistical hardness due to having the generative model, whereas the term $\|Q_k - \widehat{Q}^*\|$ represents the computational challenge and can be made desirably small after enough iterations of value iteration, and its control follows from the contraction property of ERM, which is also present in the case of known model. Similarly, in (9), the terms $\|V^{\pi_k} - \widehat{V}^{\pi_k}\|$ and $\|\widehat{V}^{\pi^*} - V^*\|$ correspond to the statistical hardness, while $\|\widehat{V}^{\pi_k} - \widehat{V}^*\|$ captures the computational hardness.

The second result concerns smoothness of Q-values under ERM when the transition function is perturbed. More specifically, it asserts how different $Q$-values of a fixed policy are in two different MDPs that differ only slightly in their transition functions. This parallels the result in (Kearns and Singh, 2002; Strehl and Littman, 2008) to ERMs.

**Lemma 3** (*Q-value smoothness under ERM*). *Consider two MDPs $M = (S, A, P, R, \gamma, \beta)$ and $\widetilde{M} = (S, A, \widetilde{P}, R, \gamma, \beta)$ differing only in their transition functions. Fix a stationary policy $\pi$, and let $Q^\pi$ and $\widetilde{Q}^\pi$ be respective Q-values of $\pi$ in $M$ and $\widetilde{M}$. It holds that $\|Q^\pi - \widetilde{Q}^\pi\| \le \xi W_1$ for $\beta < 0$, and $\|Q^\pi - \widetilde{Q}^\pi\| \le \xi W_2$ for $\beta > 0$, where $\xi := \frac{\gamma}{(1-\gamma)|\beta|} e^{|\beta|/(1-\gamma)}$ and*

$$W_1 := \max_{s,a} \Big| \sum_{s' \in \mathcal{S}} [P_{s,a}(s') - \widetilde{P}_{s,a}(s')] e^{-|\beta|(\frac{1}{1-\gamma} - V^\pi(s'))} \Big|,$$

$$W_2 := \max_{s,a} \Big| \sum_{s' \in \mathcal{S}} [P_{s,a}(s') - \widetilde{P}_{s,a}(s')] e^{-|\beta| V^\pi(s')} \Big|, \qquad with \; V^\pi(s) = \max_a Q^\pi(s, a).$$

### 6.1 Proof of Theorem 1

*Proof.* Let $\beta > 0$ and $\varepsilon > 0$. In view of the error decomposition in (8), to establish $\varepsilon$-value-correctness it suffices to ensure $\|\widehat{Q}^{\pi^*} - Q^*\| \le \varepsilon/2$ and $\|Q_k - \widehat{Q}^*\| \le \varepsilon/2$. By Lemma 2, we can have $\|Q_k - \widehat{Q}^*\| < \varepsilon/2$ using enough iterations of the optimization oracle. Further, by Lemma 3, if

$$\max_{s,a} \Big| \sum_{s' \in \mathcal{S}} [P_{s,a}(s') - \widehat{P}_{s,a}(s')] e^{-|\beta| V^*(s')} \Big| < \frac{\varepsilon(1-\gamma)|\beta|}{2\gamma} e^{-|\beta|/(1-\gamma)} =: \tau, \tag{10}$$

then $\|\widehat{Q}^{\pi^*} - Q^*\| \le \varepsilon/2$. To ensure this, we use the following lemma (proven in Appendix D):

**Lemma 4.** *Let $\pi$ be any fixed policy and $\tau > 0$. If $N > \frac{1}{2\tau^2}\big(1 - e^{-|\beta|/(1-\gamma)}\big)^2 \log(2SA/\delta)$, then it holds that*

$$(i) \qquad \max_{s,a} \Big| \sum_{s'} [P_{s,a}(s') - \widehat{P}_{s,a}(s')] e^{-\beta V^\pi(s')} \Big| < \tau, \quad with \; probability \ge 1 - \delta, \quad \beta > 0;$$

$$(ii) \qquad \max_{s,a} \Big| \sum_{s'} [P_{s,a}(s') - \widehat{P}_{s,a}(s')] e^{-|\beta|(V^\pi(s') - \frac{1}{1-\gamma})} \Big| < \tau \quad with \; probability \ge 1 - \delta, \quad \beta < 0.$$

Applying Lemma 4, inequality (i), with $\pi = \pi^\star$, we observe that (10) holds with probability at least $1 - \delta$ by picking $N \ge \frac{1}{2\tau^2}\big(1 - e^{-|\beta|/(1-\gamma)}\big)^2 \log(2SA/\delta)$. Noting that the total calls to the generative model is $T = SAN$ and substituting in the value for $\tau$, we can ensure for all $(s, a)$ that $Q_k(s, a) > Q^*(s, a) - \varepsilon$ with probability larger than $1 - \delta$ by using a total number of samples

$$T \ge \frac{2SA\gamma^2}{\varepsilon^2(1-\gamma)^2} \left( \frac{e^{|\beta|/(1-\gamma)} - 1}{|\beta|} \right)^2 \log\left( \frac{2SA}{\delta} \right).$$

The case of $\beta < 0$ is proven using very similar lines, but will use inequality (ii) in Lemma 4. $\qquad \square$

#### 6.1.1 Proof of Theorem 2

*Proof.* Let $\varepsilon > 0$ and $\beta \ne 0$. In view of the error decomposition in (9), to establish $\varepsilon$-policy-correctness it suffices to require: (i) $\|\widehat{V}^{\pi_k} - \widehat{V}^*\| \le \varepsilon/3$, (ii) $\|\widehat{V}^{\pi^*} - V^*\| \le \varepsilon/3$, and (iii) $\|V^{\pi_k} - \widehat{V}^{\pi_k}\| \le \varepsilon/3$. For (i), we

can have $\|\widehat{V}^{\pi_k} - \widehat{V}^*\| \le \varepsilon/3$ using enough iterations of the optimization oracle as a consequence of Lemma 2. To control (ii) and (iii), let us define the event

$$E := \cap_{\pi \in \Pi} \big\{ \|V^\pi - \widehat{V}^\pi\| \le \varepsilon/3 \big\}.$$

We can show that $\mathbb{P}(E) \ge 1 - \delta$ for sufficiently large $N$. Indeed, for a given $\pi \in \Pi$, an application of Lemma 3 and Lemma 4, identical to the treatment in the proof of Theorem 2, shows that $\|V^\pi - \widehat{V}^\pi\| \le \varepsilon/3$ with probability at least $1 - \delta/|\Pi|$ if $N \ge \frac{1}{2\tau^2}\big(1 - e^{-\beta/(1-\gamma)}\big)^2 \log(2SA|\Pi|/\delta)$ with $\tau = \frac{\varepsilon(1-\gamma)|\beta|}{3\gamma} e^{-|\beta|/(1-\gamma)}$. Hence,

$$\mathbb{P}(E^c) = \mathbb{P}\big(\exists \pi \in \Pi : \|V^\pi - \widehat{V}^\pi\| > \varepsilon/3\big) \le \sum_{\pi \in \Pi} \mathbb{P}\big(\|V^\pi - \widehat{V}^\pi\| > \varepsilon/3\big) \le \delta,$$

so that $\mathbb{P}(E) \ge 1 - \delta$. It is evident that conditioned on $E$, (ii) and (iii) hold. Hence, with probability greater than $1 - \delta$, $\varepsilon$-policy-correctness is maintained using a total number of model calls of

$$T_{1,\delta} = \frac{9SA\gamma^2}{\varepsilon^2(1-\gamma)^2}\left(\frac{e^{|\beta|/(1-\gamma)} - 1}{|\beta|}\right)^2 \log\left(\frac{2SA|\Pi|}{\delta}\right). \tag{11}$$

To establish the second bound, first observe that Theorem 1 implies that $\|\widehat{V}^{\pi_k} - V^*\| \le \varepsilon$ with probability exceeding $1 - \delta$ if $T \ge \frac{2SA\gamma^2}{\varepsilon^2(1-\gamma)^2|\beta|^2}\big(e^{|\beta|/(1-\gamma)} - 1\big)^2 \log(2SA/\delta)$; this is verified by noting that

$$\|\widehat{V}^{\pi_k} - V^*\| = \max_s \Big| \max_a \widehat{Q}^{\pi_k}(s,a) - \max_a Q^*(s,a)\Big| \le \max_{s,a}\big|\widehat{Q}^{\pi_k}(s,a) - Q^*(s,a)\big| = \|Q_k - Q^*\|.$$

To proceed, we make use of the following lemma, which is proven in Appendix D:

**Lemma 5.** *Let $\alpha > 0$. Let $\overline{V} \in \mathbb{R}^S$ be a value function obeying $\|V^* - \overline{V}\| < \alpha$, and $\pi^G := \operatorname{argmax}_a[R(s,a) + \gamma\rho_{s,a}(\overline{V}(s'))]$ be the greedy policy with respect to $\overline{V}$. Then, $\|V^* - V^{\pi^G}\| \le \frac{2\gamma}{1-\gamma}\alpha$.*

By construction, $\pi_k$ is the greedy policy with respect to $\widehat{V}^{\pi_k}$. Now applying Lemma 5 with $\overline{V} = \widehat{V}^{\pi_k}$, we have that

$$\|V^* - V^{\pi_k}\| \le \frac{2\gamma}{1-\gamma}\varepsilon,$$

which yields the following bound that holds with probability larger than $1 - \delta$:

$$T_{2,\delta} = \frac{8SA\gamma^4}{\varepsilon^2(1-\gamma)^4}\left(\frac{e^{|\beta|/(1-\gamma)} - 1}{|\beta|}\right)^2 \log\left(\frac{2SA}{\delta}\right). \tag{12}$$

**Final Bound.** To derive the final bound, we put together (11) and (12), while suitably adjusting the error probabilities. Therefore, $\varepsilon$-policy-correctness is guaranteed with probability exceeding $1 - \delta$ if $T \ge \min\{T_{1,\delta/2}, T_{2,\delta/2}\}$. $\qquad\square$

## 7 Sample Complexity Lower Bounds

In this section, we provide two sample complexity lower bounds. The first (Theorem 3) concerns value learning, whereas the second (Theorem 4) addresses policy learning.

**Theorem 3** (Lower bound for value learning). *There exist constants $c_1, c_2 > 0$ such that for any RL algorithm $\mathcal{U}$ that outputs a Q-value $Q^{\mathcal{U}}$, any $\delta \in (0, \frac{1}{4})$, and $\varepsilon \in \big(0, \frac{1}{40}\frac{\gamma}{|\beta|}(1 - e^{-|\beta|/(1-\gamma)})\big)$, the following holds: if the total number $T$ of samples satisfies*

$$T \le \frac{SA\gamma^2}{c_1\varepsilon^2}\frac{(e^{|\beta|/(1-\gamma)} - 3)}{|\beta|^2}\log\left(\frac{SA}{c_2\delta}\right),$$

*then there exists some MDP $M$ with $S$ states and $A$ actions for which $\mathbb{P}(\|Q_M^* - Q_T^{\mathcal{U}}\| > \varepsilon) \ge \delta$.*

**Theorem 4** (Lower bound for policy learning). *There exist constants $c_1, c_2 > 0$ such that for any RL algorithm $\mathcal{U}$ that outputs a policy $\pi_T^{\mathcal{U}}$, any $\delta \in (0, \frac{1}{4})$, and $\varepsilon \in \left(0, \frac{1}{50} \frac{\gamma}{|\beta|}(1 - e^{-|\beta|/(1-\gamma)})\right)$, it holds that if the total number $T$ of samples satisfies*

$$T \leq \frac{SA\gamma^2}{c_1\varepsilon^2} \frac{(e^{|\beta|/(1-\gamma)} - 3)}{|\beta|^2} \log\left(\frac{S}{c_2\delta}\right),$$

*then there exists some MDP $M$ with $S$ states and $A$ actions for which $\mathbb{P}(\|V_M^* - V^{\pi_T^{\mathcal{U}}}\| > \varepsilon) \geq \delta$.*

While analogous policy learning lower bounds are often stated in the risk-neutral literature, explicit proofs are typically omitted, to the best of our knowledge. For completeness, we provide a detailed, step-by-step derivation, emphasizing its close connection to the corresponding value-learning lower bound as well as the subtle differences that arise in the final guarantee.

Theorems 3–4 establish that an exponential dependence on the effective horizon $1/(1 - \gamma)$ in the sample complexity is unavoidable under both value and policy learning. These bounds cover both risk-averse ($\beta > 0$) and risk-seeking ($\beta < 0$) agents, providing strong impossibility results for recursive ERM. Comparing with the sample complexity bounds of MB-ERM-QVI (Theorems 1–2), we observe a similar exponential dependence; however, a gap of order $\frac{1}{(1-\gamma)^2}e^{|\beta|/(1-\gamma)}$ remains. Closing this gap may require either a refined analysis of MB-ERM-QVI or more sophisticated algorithmic ideas. Nevertheless, these lower bounds confirm that risk-sensitive RL under ERM is fundamentally harder in the worst case than the risk-neutral setting, where minimax sample complexity scales polynomially with $1/(1 - \gamma)$.

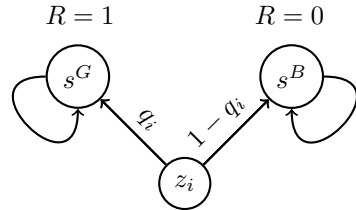

Figure 1: Hard-to-learn MDP construction

**Proof sketch.** The proofs are provided in Appendix F; here we sketch the main ideas for value learning (Theorem 3). The construction involves a class of hard-to-learn MDPs (Figure 1) with two absorbing states $s^G$ and $s^B$, yielding rewards $R = 1$ and $R = 0$ under any action, respectively. All other state-action pairs $z$ give zero reward and transition only to $s^G$ or $s^B$ with probability $P(s^G|z) = q$ and $P(s^B|z) = 1 - q$, for some $q > 0$. This construction critically allows us to calculate explicitly $Q^*(z)$ for a given parameter $q$ and for two different MDPs $M_0, M_1$ in the class, where $q_0 = p$ and $q_1 = p + \alpha$ for appropriately chosen values of $p$ and $\alpha$. It is key to choose them in a way to ensure that $Q_{M_1}^*(z) - Q_{M_0}^*(z) > 2\varepsilon$, which means that any specific algorithmic output $Q^{\mathcal{U}}(z)$ cannot be $\varepsilon$-close to both $Q_{M_1}^*(z)$ and $Q_{M_0}^*(z)$. We then show by a likelihood ratio argument that any algorithm $\mathcal{U}$ that is $(\varepsilon, \delta)$-correct on $M_0$, i.e. that $\mathbb{P}_0(|Q_{M_0}^*(z) - Q^{\mathcal{U}}(z)| \leq \varepsilon) \geq 1 - \delta$, will also satisfy that $\mathbb{P}_1(|Q_{M_0}^*(z) - Q^{\mathcal{U}}(z)| \leq \varepsilon) > \delta$ provided that the algorithm does not try out $z$ enough times on $M_0$ and exactly because $Q_{M_1}^*(z) - Q_{M_0}^*(z) > 2\varepsilon$, the event $\{|Q_{M_0}^*(z) - Q^{\mathcal{U}}(z)| \leq \varepsilon\}$ is disjoint from the event on being $\varepsilon$-close to $Q_{M_1}^*$. The final part of the proof is to exploit that the different state-action pairs contain no information about each other, which allows for an independence argument for the estimation of $Q^{\mathcal{U}}(z)$ and $Q^{\mathcal{U}}(z')$ for $z \neq z'$.

We note that, in the course of this analysis, we also correct a minor issue in Lemma 17 of Gheshlaghi Azar et al. (2013). Specifically, the issue arises in the derivation of a lower bound on the likelihood ratio between two Bernoulli distributions with means $p$ and $p + \alpha$ on a high probability event, for $p \geq \frac{1}{2}$. Additionally, we establish a corresponding bound for $p < \frac{1}{2}$, which may be of independent interest. For policy learning, a similar construction is used, augmented with a known action $a_0$ to facilitate the analysis.

**Algorithmic intuition.** The above construction also provides insight into how a model-based algorithm such as MB-ERM-QVI behaves on these instances. Since all rewards are zero except in the absorbing states,

learning reduces to estimating the transition probability $q = P(s^G|z)$ for each state-action pair $z$. The algorithm therefore forms an empirical estimate $\hat{q}$ and computes $Q$-values based on this estimate. On the hard instances $M_0$ and $M_1$, the true probabilities differ only by a small amount ($q_0 = p$ vs. $q_1 = p + \alpha$). When the number of samples is limited, the empirical estimate $\hat{q}$ will typically not be accurate enough to reliably distinguish between these two cases. As a result, the algorithm may construct a model that is statistically consistent with both $M_0$ and $M_1$, leading to $Q$-value estimates that are necessarily inaccurate for at least one of them.

**Remark 4.** *The best lower bound in the risk-neutral setting is derived in (Gheshlaghi Azar et al., 2013) using a richer construction than above. However, with a risk-sensitive learning objective, the optimal Q-value function in that construction does not admit an analytical solution, which is crucial for tuning transition probabilities and deriving our bounds.*

**Remark 5.** *We note that the bound becomes vacuous for $|\beta| \leq (1-\gamma)\log(3)$. This is partly due to a final approximation introduced to make the bound more interpretable; importantly, this approximation does not affect the dependence on $|\beta|$ for large $|\beta|$. Even without this approximation, the bound still becomes vacuous as $\beta \to 0$. This behavior arises because, in this limit, $p \to 1$ or $p \to 0$ depending on the direction of the limit. Since our information-theoretic argument yields a sample complexity proportional to $p(1-p)$, the bound vanishes in this regime.*

# 8 Numerical Experiments

We conduct a numerical experiment to showcase the empirical performance of MB-ERM-QVI. We consider a RiverSwim MDP (Strehl and Littman, 2008) (Figure 2a) with 8 states and discount factor $\gamma = 0.95$. In this MDP, there are two actions in each state, corresponding to moving 'left' or 'right'. The rewards are zero, except in two places: a reward of 0.05 (low reward) under 'left' in the left-most state ($s = 1$), and a reward of 1 (high reward) under 'right' in the right-most state ($s = L$). The low reward is easy to access because of the actions with deterministic transitions. The agent has a risk-sensitive objective defined using recursive ERM. To showcase the performance of MB-ERM-QVI, we consider three different values of $\beta \in \{0, 1, 1.25\}$; we recall that $\beta = 0$ corresponds to the risk-neutral case. For each value of $\beta$, we generated datasets of increasing sizes $T \in \{160, 320, \dots, 1600\}$ (multiples of $SA = 16$) by sampling each state-action pair in the MDP. The MB-ERM-QVI algorithm was run on each dataset to produce an output policy $\hat{\pi}$. This procedure was run for 1000 independent runs.

Figure 2 depicts $\|V^* - V^{\hat{\pi}}\|$ averaged over the 1000 runs for the three values of $\beta$. The true value of $\hat{\pi}$ is computed using a value iteration procedure based on (7). In this figure, one may consider a particular value of $\varepsilon$ to observe the number of samples needs to output an $\varepsilon$-optimal policy. It is evident that this number of samples increases as $\beta$ increases. Furthermore, it is evident that even a small increase in $\beta$ (from 1 to 1.25) leads to a large increase in the number of samples to learn an $\varepsilon$-optimal policy. We also note that when $\hat{\pi}$ matches the optimal policy for all the runs, the curve would hit the horizontal axis. These results demonstrate that learning a near-optimal policy under ERM could be substantially more demanding in terms of data even in such a rather simple MDP, and the data efficiency may be severely impacted as the risk parameter $\beta$ increases.

# 9 Concluding Remarks

We studied sample complexities of value learning and policy learning in finite discounted MDPs, where the agent exhibits recursive risk preferences modeled via the ERM and has access to a generative model. The generative model setting is commonly used in theoretical RL as it provides a clean framework in which the statistical difficulty of the problem can be isolated and precisely characterized. In particular, it removes effects due to exploration and data collection, allowing us to focus on the intrinsic difficulty induced by the risk-sensitive objective. We introduced a model-based algorithm, MB-ERM-QVI, and derived PAC-type bounds on its sample complexity. In addition, we established sample complexity lower bounds for both policy and value learning. These lower bounds reveal that an exponential dependence on the horizon $1/(1-\gamma)$ is unavoidable in the worst case, demonstrating that this setting is fundamentally harder than the risk-neutral

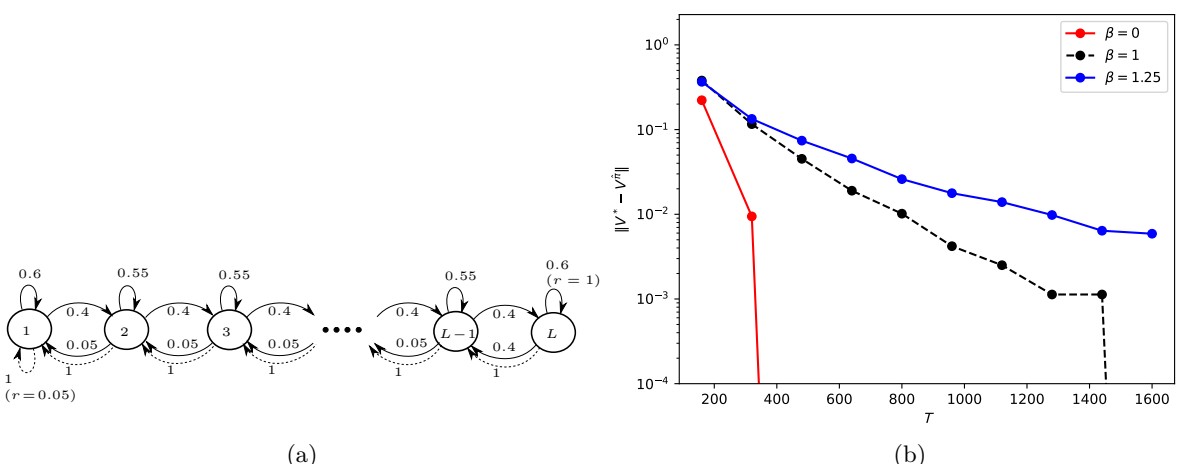

(a)

(b)

Figure 2: (a) The $L$-state RiverSwim MDP (Strehl and Littman, 2008); (b) Policy learning error $\|V^* - V^{\hat{\pi}}\|$ under MB-ERM-QVI.

case. To our knowledge, these constitute the first upper and lower bounds for this setting. The bounds are tight in $S$, $A$, $\delta$, and $\varepsilon$, but gaps remain in the dependence on $1/(1-\gamma)$.

Closing these remaining gaps leads to interesting directions for future work. It is not immediately clear whether this requires improving the upper bound, the lower bound, or both. The plug-in estimator used in MB-ERM-QVI is provably optimal in the risk-neutral case, raising the natural question of whether this optimality extends to ERM. On the other hand, designing hard MDP instances is subtle as they should be difficult to learn, yet still tractable enough to compute value functions; this is a challenge amplified in the risk-sensitive setting. Moreover, our lower bound is valid for $|\beta| > (1-\gamma)\log(3)$, and it would therefore be interesting to derive a corresponding result that recovers the risk-neutral lower bound as $\beta$ tends to zero.

Another promising direction is the development of model-free algorithms for this setting and the analysis of their statistical efficiency. It would also be valuable to extend the study to MDPs with function approximation (e.g., Zhou et al. (2021)), as well as to more complex RL settings, including offline RL (Rashidinejad et al., 2021), where data is collected under a fixed (but unknown) behavior policy, and online RL (Strehl and Littman, 2008; Lattimore and Hutter, 2014), where the agent's learned policy directly impacts the data collection process. For extensions to offline RL, one may adopt a model-based approach, wherein the plug-in estimators combined with pessimism-style corrections are often used (e.g., Li et al. (2024b)); some of the techniques developed in this paper may prove useful in analyzing such methods.

### Acknowledgments

The authors would like to acknowledge the support from Independent Research Fund Denmark, grant number 1026-00397B.

### Conflicts of Interests

The authors claim no conflict of interests.

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

## A   Risk Measures

In this section, we give a very brief introduction to risk measures. In the actuarial and mathematical finance literature, working with both losses and rewards is common. A good introduction is (Föllmer and Schied, 2010), which like us uses the rewards formulation. Due to this ambiguity in the literature, we here collect some precise definitions for the reward setting, and then list some of the most important risk measures.

To begin with, let $(\Omega, \mathcal{F}, \mathbb{P})$ be a background probability space, and $\mathcal{M}$ some convex cone of random variables defined on the background space. That is, for any $X, Y \in \mathcal{M}$ and $\lambda > 0$, it holds that $X + Y \in \mathcal{M}$ and $\lambda X \in \mathcal{M}$.

**Definition 2** (Risk measure). *A functional $\psi : \mathcal{M} \to \mathbb{R}$ is said to be a* risk measure *if it satisfies the following properties:*

$$\psi(0) = 0, \qquad\qquad\qquad (Normalization)$$
$$\text{if } X \leq Y \text{ then } \psi(X) \geq \psi(Y), \qquad\qquad\qquad (Monotonicity)$$
$$\psi(X + c) = \psi(X) - c, \quad \forall c \in \mathbb{R}. \qquad\qquad (Translation\ invariance)$$

*If, in addition, $\psi$ satisfies the properties*

$$\psi(cX) = c\psi(X), \quad \forall c > 0, \qquad\qquad (Positive\ homogeneity)$$
$$\psi(X + Y) \leq \psi(X) + \psi(Y), \qquad\qquad (Sub\text{-}additivity)$$

*it is called a* coherent risk measure. *A weaker notion is* convex risk measure, *which is one obeying*

$$\psi(\lambda X + (1 - \lambda)Y) \leq \lambda\psi(X) + (1 - \lambda)\psi(Y), \quad \forall \lambda \in [0, 1]. \qquad (Convexity)$$

*Finally, a risk-measure $\psi$ is called* law-invariant *if $\psi(X)$ only depends on the distribution of $X$ under $\mathbb{P}$.*

We now mention some examples of risk measures.

**Entropic Risk Measure (ERM).**   The risk measure given by

$$\text{ERM}_\beta(X) = \frac{1}{\beta} \log\left(\mathbb{E}[e^{-\beta X}]\right)$$

is known as the entropic risk measure (ERM) with parameter $\beta \neq 0$. Notably, ERM is not coherent (see, e.g., Föllmer and Schied (2010)) as it is does not satisfy the positive homogeneity property. Letting $\beta \to 0$ one recovers the negative expectation $-\mathbb{E}[X]$, and letting $\beta \to \infty$ yields the negative essential infimum risk measure.

**Value-at-Risk (VaR).**    The risk measure given by

$$\text{VaR}_\alpha(X) := q_\alpha(X) := \inf\{x \in \mathbb{R} : F_X(x) \geq \alpha\}$$

is called the Value-at-Risk (VaR) at level $\alpha \in (0,1)$. VaR is in general not sub-additive, and hence also not coherent.

**Conditional Value-at-Risk (CVaR).**    The risk measure given by

$$\text{CVaR}_\alpha(X) := \frac{1}{\alpha} \int_0^\alpha \text{VaR}_u(X) du$$

is known as the Conditional Value-at-Risk (CVaR), or sometimes as the expected shortfall (ES). It is known to be a coherent risk-measure.

All the examples so far are evidently law-invariant.

It is worth highlighting that the actual functional $\rho$ used to rank random variables is the *negative* of the ERM-risk measure $\text{ERM}_\beta(X)$, introduced above, with the interpretation being that a lower quantity of risk is preferable. More formally, we consider the functional $\rho : \mathcal{M} \to \mathbb{R}$ given by $\rho(X) := -\text{ERM}_\beta(X)$, featuring the following properties:

$$\rho(0) = 0, \qquad \qquad \text{(Normalization)}$$
$$\text{if } X \leq Y \text{ then } \rho(X) \leq \rho(Y), \qquad \qquad \text{(Monotonicity)}$$
$$\rho(X + c) = \rho(X) + c. \qquad \qquad \text{(Translation invariance)}$$

It is common in the literature to overload notation and also refer to $\rho$ as the ERM and we will do so and henceforth we will no longer work directly with risk measures, but only with this specific functional $\rho$. It follows immediately from the normalization and translation invariance that $\rho(c) = c$ for any $c \in \mathbb{R}$.

We will often use the short-hand notation $\rho_{s,a}(V(s'))$ as $\rho$ applied to the random variable $X$ that takes on the values $\{V(s')\}_{s' \in S}$ with probabilities $\mathbb{P}(X = V(s')) = P(s'|s,a)$.

# B    Bellman Optimality and Bellman Recursions

In this section, we properly define the state-value functions and state-action value functions of any possibly history-dependent policy $\pi$, and show that the problem of finding an optimal policy can be achieved by a stationary policy and that the value functions satisfy Bellman recursions when the value functions are defined iteratively with respect to ERM. Several similar results exist in the literature, e.g., (Asienkiewicz and Jaśkiewicz, 2017) and (Bäuerle and Jaśkiewicz, 2024) that cover the case of $\beta > 0$. These results are derived under more general assumptions on $\mathcal{S}$ and $\mathcal{A}$. These general assumptions are trivially satisfied when $\mathcal{S}$ and $\mathcal{A}$ are finite but their proofs require assumptions on the functionals to ensure the existence of a stationary optimal policy usually by invoking a measurable selection theorem. We avoid this complication by only considering finite $\mathcal{S}$ and $\mathcal{A}$, and we in turn give the first proof for state-action value functions and not just for value-functions, which is needed as we consider the problem of learning.

Let $M = (\mathcal{S}, \mathcal{A}, P, R, \gamma, \rho)$ be a finite MDP with $\rho$ being ERM, and $R(s,a) \in [0,1]$ a deterministic reward function. Let $D = \mathcal{S} \times \mathcal{A}$, $H_1 = \mathcal{S}$, and $H_k = D^{k-1} \times \mathcal{S}$ for $k \geq 2$ the set of all possible histories up to stage $k$. A policy $\pi = (\pi_k)_{k \in \mathbb{N}}$ is a sequence of maps $\pi_k : H_k \to \mathcal{A}$. We denote the set of all policies by $\Pi$, and identify the set of all stationary policies with the set of measurable maps $F$ from $\mathcal{S}$ to $\mathcal{A}$, which is simply the set of all maps from $\mathcal{S}$ to $\mathcal{A}$ since all maps between finite sets that are both equipped with the discrete topology are measurable with respect to the induced Borel $\sigma$-algebras. Let $B(H_k)$ be the set of maps $V_k : H_k \to \mathbb{R}$ equipped with the supremum norm, and let $\pi = (\pi_k)_{k \in \mathbb{N}}$ be any policy. For any $V_{k+1} \in B(H_{k+1})$ and $h_k \in H_k$, we denote by $\rho_{h_k,\pi_k}(V_{k+1})$ the functional $\rho$ applied to the random variable concentrated on the set $\{V_{k+1}(h_k, \pi_k(h_k), s')\}_{s' \in S}$ with $\mathbb{P}(s_{k+1} = s') = \mathbb{P}(s'|s_k, \pi_k(h_k))$. By monotonicity of $\rho$, we get that $\rho_{h_k,\pi_k}(V_{k+1}) \leq \|V_{k+1}\|$.

Next, we define the operators $L_{\pi_k} : B(H_{k+1}) \to B(H_k)$ by

$$(L_{\pi_k} V_{k+1})(h_k) = L_{\pi_k, V_{k+1}}(h_k) := R(s_k, \pi_k(h_k)) + \gamma \rho_{h_k, \pi_k}(V_{k+1}).$$

Similarly, we define $L_a : B(H_{k+1}) \to B(H_k)$ by

$$(L_a V_{k+1})(h_k) = L_{a, V_{k+1}}(h_k) := R(s_k, a) + \gamma \rho_{s_k, a}(V_{k+1}),$$

with $\rho_{s_k, a}$ defined analogously as $\rho_{h_k, \pi_k}$ as above. By the basic properties of risk-measures, it follows directly that $0 \le L_{\pi_k, V_{k+1}}(h_k) \le 1 + \gamma \|V_{k+1}\|$ and similarly for $L_a$.

For any initial state $s_0 = s$, we define the $N$-step discounted utility as

$$J_N(s, a, \pi) := (L_a \circ L_{\pi_2} \circ \cdots \circ L_{\pi_N}) \mathbf{0}(s)$$

where $\mathbf{0}(h_k) = 0$ for all $h_k \in H_k$ and all $k \in \mathbb{N}$.

By monotonicity of $\rho$, it holds that the sequence $(J_N(s, a, \pi))_{N \in \mathbb{N}}$ is non-decreasing and bounded in the interval $[0, \frac{1}{1-\gamma}]$ for any $(s, a, \pi) \in \mathcal{S} \times \mathcal{A} \times \Pi$, and so the limit

$$J(s, a, \pi) := \lim_{N \to \infty} J_N(s, a, \pi)$$

exists for any state $s$, any action $a$, and any policy $\pi$.

The agent wishes to find $J^*(s, a) = \sup_{\pi \in \Pi} J(s, a, \pi)$ and an optimal policy $\pi^*$ attaining $J^*(s, a)$, namely, $J(s, a, \pi^*) = J^*(s, a)$.

**Theorem 5.** *There exist a unique non-negative function $Q \in B(\mathcal{S} \times \mathcal{A})$ (non-negative map from $\mathcal{S} \times \mathcal{A} \to \infty$ equipped with the sup-norm) and a stationary decision rule $f^* : \mathcal{S} \to \mathcal{A}$ such that*

$$Q(s, a) = R(s, a) + \gamma \rho_{s,a}\left(\max_{a'} Q(s', a')\right),$$
$$= R(s, a) + \gamma \rho_{s,a}(Q(s', f^*(s'))).$$

*Moreover, $Q(s, a) = J^*(s, a) = J(s, a, f^*)$ meaning that $f^*$ is an optimal stationary policy.*

*Proof.* We start by proving existence of $Q$. Let $L : B(\mathcal{S} \times \mathcal{A}) \to B(\mathcal{S} \times \mathcal{A})$ denote the operator given by

$$LQ(s, a) := R(s, a) + \gamma \rho_{s,a}\left(\max_{a'} Q(s', a')\right).$$

Let $Q_1, Q_2 \in B(\mathcal{S} \times \mathcal{A})$. We then have for all $(s, a)$ that

$$LQ_1(s, a) - LQ_2(s, a) = \gamma[\rho_{s,a}(\max_{a'} Q_1(s', a')) - \rho_{s,a}(\max_{a'} Q_2(s', a'))]$$
$$= \gamma[\rho_{s,a}(\max_{a'} Q_1(s', a')) - \rho_{s,a}(\max_{a'} Q_1(s', a') - \max_{a'} Q_1(s', a') + \max_{a'} Q_2(s', a'))]$$
$$\le \gamma[\rho_{s,a}(\max_{a'} Q_1(s', a')) - \rho_{s,a}(\max_{a'} Q_1(s', a') + \max_{a'}\{Q_2(s', a') - Q_1(s', a')\})]$$
$$\le \gamma[\rho_{s,a}(\max_{a'} Q_1(s', a')) - \rho_{s,a}(\max_{a'} Q_1(s', a') + \|Q_1 - Q_2\|)]$$
$$= \gamma \|Q_1 - Q_2\|.$$

We start by showing that $L : B(\mathcal{S} \times \mathcal{A}) \to B(\mathcal{S} \times \mathcal{A})$, that is, it takes non-negative functions and returns non-negative functions. By normalization and monotonicity of $\rho$, we have for any non-negative $Q \in B(\mathcal{S} \times \mathcal{A})$ that

$$LQ(s, a) = R(s, a) + \gamma \rho_{s,a}(\max_{a'} Q(s, a)) \ge 0 + \gamma \rho_{s,a}(0) = 0.$$

By a completely similar argument, we have $LQ_2(s, a) - LQ_1(s, a) \le \gamma \|Q_1 - Q_2\|$, so that $L$ is a contraction, and since for $\mathcal{S} \times \mathcal{A}$ we can identify $B(\mathcal{S} \times \mathcal{A})$ with the closed subset of the complete metric space $(\mathbb{R}^{S \times A}, \|\cdot\|)$

that consists of vectors with non-negative coordinates. Since this subspace is closed, it is also a complete metric space and the existence of $Q$ then follows from the Banach fixed-point theorem.

Since there are only finitely many states and actions, we can pick a stationary decision rule where $f^*(s)$ is an arbitrary element of $\mathrm{argmax}_a\, Q(s,a)$.

Let $V$ be the function given by $V(s) := Q(s, f^*(s))$ for all $s$. We then observe that

$$V(s) \geq R(s,a) + \gamma \rho_{s,a}(V(s'))$$

for every $s \in \mathcal{S}$. Let $\pi = (\pi_k)_k \in \mathbb{N}$ be any policy in $\Pi$. The above inequality then shows that for any history $h_k, k \in \mathbb{N}$, we have that $V(s_k) \geq L_{\pi_k} V(h_k)$ and furthermore we note that $Q(s_1, a) = L_a V(h_1)$. This implies for any $N \in \mathbb{N}$ that

$$Q(s,a) \geq (L_a \circ L_{\pi_2} \circ \cdots \circ L_{\pi_N})V(s) \geq (L_a \circ L_{\pi_2} \circ \cdots \circ L_{\pi_N})\mathbf{0}(s) = J_N(s,a,\pi),$$

where we have used that $Q(s,a) \geq 0$. Finally, taking the limit, we find that $Q(s,a) \geq J(s,a,\pi)$.

Finally, we aim to show that $Q(s,a) \leq J(s,a,f^*)$. By induction, we wish to show that $V(s) \leq J_N(s, f^*(s), f^*) + \gamma^N \|V\|$ for all $N \in \mathbb{N}$. For the induction step, we start by noting that $J_1(s, f^*(s), f^*) = R(s, f^*(s))$ and so

$$\begin{aligned}
V(s) &= R(s, f^*(s)) + \gamma \rho_{s, f^*(s)}(V(s')) \\
&\leq R(s, f^*(s)) + \gamma \rho_{s, f^*(s)}(\|V\|) \\
&= R(s, f^*(s)) + \gamma \|V\| \\
&= J_1(s, f^*(s), f^*) + \gamma \|V\|,
\end{aligned}$$

for all $(s,a) \in \mathcal{S} \times \mathcal{A}$. For the induction step, we assume that $V(s) \leq J_N(s, f^*(s), f^*) + \gamma^N \|V\|$. By using that $V(s) = L_{f^*} V(s)$ and that $L$ is monotone, we see that

$$\begin{aligned}
V(s) &= L_{f^*} V(s) \\
&\leq L_{f^*}(J_N(s, f^*(s), f^*) + \gamma^N \|V\|) \\
&= \left(R(\cdot, f^*(\cdot)) + \gamma \rho_{\cdot, f^*(\cdot)}(J_N(\cdot, f^*(\cdot), f^*) + \gamma^N \|V\|)\right)(s) \\
&= J_{N+1}(s, f^*(s), f^*) + \gamma^{N+1} \|V\|,
\end{aligned}$$

from which taking the limit as $N \to \infty$, we get that $V(s) \leq J(s, f^*(s), f^*)$.

Finally, since

$$Q(s,a) = L_a V(s) \leq L_a J(s, f^*(s), f^*) = J(s,a,f^*),$$

the conclusion holds.

Since this shows that an optimal stationary policy exists, it will suffice to consider only stationary policies and one can by completely analogous arguments show that for any stationary policy $\pi$, there exists a non-negative map $Q^\pi \in B(\mathcal{S} \times \mathcal{A})$ such that $Q^\pi(s,a) = J(s,a,\pi)$, that is, $Q^\pi$ satisfies the Bellman recursion:

$$Q^\pi(s,a) = R(s,a) + \gamma \rho_{s,a}(Q^\pi(s', \pi(s'))),$$

and similarly for state-value functions $V^\pi(s) := Q^\pi(s, \pi(s))$. $\qquad\square$

We also remark that in the proof, we see directly that $Q(s,a) \in [0, \frac{1}{1-\gamma}]$ for all $(s,a)$.

## C  Technical Lemmas

Recall that $Q_k$ is the Q-function output by the algorithm after $k$ iterations, $\pi_k$ is the greedy policy with respect to $Q_k$, and that $\pi^*$ is an optimal policy of the true MDP $M$.

The first lemma establishes a decomposition result for MB-ERM-QVI, whose proof follows very similar lines to the proof of Lemma 3 in (Agarwal et al., 2020).

**Lemma 6.** *For any state-action pair $(s, a) \in \mathcal{S} \times \mathcal{A}$,*

$$Q_k(s, a) \geq Q^*(s, a) - \|Q_k - \widehat{Q}^*\| - \|\widehat{Q}^{\pi^*} - Q^*\|.$$

*Further, for any state $s \in \mathcal{S}$,*

$$V^{\pi_k}(s) \geq V^*(s) - \|V^{\pi_k} - \widehat{V}^{\pi_k}\| - \|\widehat{V}^{\pi_k} - \widehat{V}^*\| - \|\widehat{V}^{\pi^*} - V^*\|.$$

*Proof.* For any $(s, a) \in \mathcal{S} \times \mathcal{A}$, we have

$$\begin{aligned} Q_k(s, a) - Q^*(s, a) &= Q_k(s, a) - \widehat{Q}^*(s, a) + \widehat{Q}^*(s, a) - Q^*(s, a) \\ &\geq Q_k(s, a) - \widehat{Q}^*(s, a) + \widehat{Q}^{\pi^*}(s, a) - Q^*(s, a) \\ &\geq -\|Q_k - \widehat{Q}^*\| - \|\widehat{Q}^{\pi^*} - Q^*\|. \end{aligned}$$

Similarly, for any $s \in \mathcal{S}$, we have

$$\begin{aligned} V^{\pi_k}(s) - V^*(s) &= V^{\pi_k}(s) - \widehat{V}^{\pi_k}(s) + \widehat{V}^{\pi_k}(s) - \widehat{V}^*(s) + \widehat{V}^*(s) - V^*(s) \\ &\geq V^{\pi_k}(s) - \widehat{V}^{\pi_k}(s) + \widehat{V}^{\pi_k}(s) - \widehat{V}^*(s) + \widehat{V}^{\pi^*}(s) - V^*(s) \\ &\geq -\|V^{\pi_k} - \widehat{V}^{\pi_k}\| - \|\widehat{V}^{\pi_k} - \widehat{V}^*\| - \|\widehat{V}^{\pi^*} - V^*\|, \end{aligned}$$

and the lemma follows. $\qquad\square$

Next, we present two lemmas that collect a few useful inequalities. Some of these may be standard results, but for concreteness, we collect them here.

**Lemma 7.** *It holds that*

$$\begin{aligned} \log(1 - x) &\geq -x - x^2 + x^3, & \forall x \in [0, \tfrac{1}{5}] \\ \log(1 - x) &\geq -x - x^2, & \forall x \in [0, \tfrac{1}{2}] \\ \log(1 + x) &\geq x - x^2, & \forall x \in [0, \infty) \\ \log(1 + x) &\geq \frac{x}{2}, & \forall x \in [0, 1]. \end{aligned}$$

*Proof.* We only prove the first claim, as the rest could be proven using the same technique after some elementary calculations.

Let $f(x) = \log(1 - x)$ and $g(x) = -x - x^2 + x^3$. It holds that $f(0) = g(0)$, and since we have $f'(x) = -\frac{1}{1-x}$ and $g'(x) = -1 - 2x + 3x^2$, it follows easily that

$$f'(x) \geq g'(x) \Leftrightarrow 0 \leq x(1 - 5x + 3x^2),$$

where the inequality is satisfied for all $x \in [0, \frac{5 - \sqrt{13}}{6}] \subseteq [0, \frac{1}{5}]$. The result then follows from the fundamental theorem of calculus. $\qquad\square$

**Lemma 8.** *Let $\alpha > 1$. For any $x \in [0, \frac{1}{\alpha}]$, it holds that*

$$1 - (1 - x)^\alpha \geq \frac{x\alpha}{2}.$$

*Proof.* Define $f(x) = 1 - (1 - x)^\alpha - \frac{x\alpha}{2}$. Since $f''(x) = -\alpha(\alpha - 1)(1 - x)^{\alpha - 2} < 0$, $f$ is strictly concave. Further, since $f(0) = 0$ and $f(\frac{1}{\alpha}) = \frac{1}{2}(1 - \frac{1}{\alpha})^\alpha > \frac{1}{2} - \frac{1}{e} > 0$, $f$ is positive on the interval $[0, \frac{1}{\alpha}]$, and the result follows. $\qquad\square$

# D    Analysis of MB-ERM-QVI: Missing Proofs

## D.1    Proof of Lemma 1

*Proof.* We only give the proof for $\mathcal{T}$ as the claim for $\mathcal{T}^\pi$ could be proven using extremely similar lines.

Consider two maps $Q : \mathbb{R}^{S \times A} \to \mathbb{R}^{S \times A}$ and $W : \mathbb{R}^{S \times A} \to \mathbb{R}^{S \times A}$, and let $Q' = \mathcal{T}Q$ and $W' = \mathcal{T}W$ be their respective $\mathcal{T}$-transforms. Let $(s, a)$ be any pair such that $|Q'(s, a) - W'(s, a)| = \|Q' - W'\|_\infty$, and assume without loss of generality that $Q'(s, a) \geq W'(s, a)$. Further, define

$$V(s) := \max_a Q(s, a), \qquad X(s) := \max_a W(s, a).$$

Assuming $\beta > 0$, we then have

$$
\begin{aligned}
\|Q' - W'\| &= Q'(s, a) - W'(s, a) \\
&= -\frac{\gamma}{\beta} \log \left( \sum_{s'} P(s'|s, a) e^{-\beta V(s')} \right) + \frac{\gamma}{\beta} \log \left( \sum_{s'} P(s'|s, a) e^{-\beta X(s')} \right) \\
&= -\frac{\gamma}{\beta} \log \left( \sum_{s'} P(s'|s, a) e^{-\beta X(s') - \beta(V(s') - X(s'))} \right) + \frac{\gamma}{\beta} \log \left( \sum_{s'} P(s'|s, a) e^{-\beta X(s')} \right) \\
&\leq -\frac{\gamma}{\beta} \log \left( \sum_{s'} P(s'|s, a) e^{-\beta X(s') - \beta\|V - X\|} \right) + \frac{\gamma}{\beta} \log \left( \sum_{s'} P(s'|s, a) e^{-\beta X(s')} \right) \\
&= \gamma \|V - X\| \\
&\leq \gamma \|Q - W\|,
\end{aligned}
$$

and the lemma follows. The proof for the case of $\beta < 0$ follows very similar lines, and is thus omitted. $\qquad\square$

## D.2    Proof of Lemma 2

*Proof.* By Lemma 1, we have that $\mathcal{T}$ is a $\gamma$-contraction and that $Q^*$ is its unique fixed point. We thus have $\|Q_k - Q^*\| = \|\mathcal{T}Q_{k-1} - \mathcal{T}Q^*\| \leq \gamma \|Q_{k-1} - Q^*\|$. Applying this inequality $k$ times yields

$$\|Q_k - Q^*\| \leq \gamma^k \|Q_0 - Q^*\| \leq \frac{\gamma^k}{1 - \gamma}.$$

Solving $\frac{\gamma^k}{1-\gamma}$ for $k$, we get that if $k > \frac{-\log((1-\gamma)\varepsilon)}{\log(1/\gamma)}$, then $\|Q_k - Q^*\| < \varepsilon$, thus proving the first claim.

To show the other claim, we start by noting that $\|V^{\pi_k} - V^*\| \leq \|Q^{\pi_k} - Q^*\|$. Furthermore, by design we have that $\mathcal{T}^{\pi_k} Q^{\pi_k} = Q^{\pi_k}$ and that $\mathcal{T}Q_k = \mathcal{T}^{\pi_k} Q_k$. Thus,

$$\|Q^{\pi_k} - Q^*\| \leq \|Q^{\pi_k} - Q_k\| + \|Q_k - Q^*\|.$$

The first term in the right-hand side is bounded as follows:

$$
\begin{aligned}
\|Q^{\pi_k} - Q_k\| &= \|\mathcal{T}^{\pi_k} Q^{\pi_k} - Q_k\| \\
&\leq \|\mathcal{T}^{\pi_k} Q^{\pi_k} - \mathcal{T}Q_k\| + \|\mathcal{T}Q_k - Q_k\| \\
&= \|\mathcal{T}^{\pi_k} Q^{\pi_k} - \mathcal{T}^{\pi_k} Q_k\| + \|\mathcal{T}Q_k - \mathcal{T}Q_{k-1}\| \\
&\leq \gamma \|Q^{\pi_k} - Q_k\| + \gamma \|Q_k - Q_{k-1}\|,
\end{aligned}
$$

which means that

$$\|Q^{\pi_k} - Q_k\| \leq \frac{\gamma}{1 - \gamma} \|Q_k - Q_{k-1}\| \leq \frac{\gamma^k}{1 - \gamma} \|Q_1 - Q_0\| \leq \frac{\gamma^k}{(1 - \gamma)^2}.$$

The proof is completed by observing that picking $k > \log \left( \frac{2}{(1-\gamma)^2 \varepsilon} \right) / \log(1/\gamma)$ implies $\|V^{\pi_k} - V^*\| < \varepsilon$. $\qquad\square$

### D.3 Proof of Lemma 3

*Proof.* Let $(s, a)$ be any pair such that $|Q_1(s, a) - Q_2(s, a)| = \|Q_1 - Q_2\|_\infty$. There are four cases to consider, corresponding to the combinations of $\beta > 0$ or $\beta < 0$, and whether $Q_1(s, a) > Q_2(s, a)$ or $Q_2(s, a) > Q_1(s, a)$.

**Case 1: $\beta > 0$ and $Q_1(s, a) > Q_2(s, a)$.** We have

$$
\begin{aligned}
\|Q_1 - Q_2\| &= \frac{\gamma}{\beta} \log \left( \frac{\sum_{s'} P_2(s'|s, a) e^{-\beta V_2(s')}}{\sum_{s'} P_1(s'|s, a) e^{-\beta V_1(s')}} \right) \\
&= \frac{\gamma}{\beta} \log \left( \frac{\sum_{s'} P_2(s'|s, a) e^{-\beta V_1(s') - \beta(V_2(s') - V_1(s'))}}{\sum_{s'} P_1(s'|s, a) e^{-\beta V_1(s')}} \right) \\
&\leq \frac{\gamma}{\beta} \log \left( e^{\beta \|V_1 - V_2\|} \frac{\sum_{s'} P_2(s'|s, a) e^{-\beta V_1(s')}}{\sum_{s'} P_1(s'|s, a) e^{-\beta V_1(s')}} \right) \\
&= \gamma \|V_1 - V_2\| + \frac{\gamma}{\beta} \log \left( \frac{\sum_{s'} P_2(s'|s, a) e^{-\beta V_1(s')}}{\sum_{s'} P_1(s'|s, a) e^{-\beta V_1(s')}} \right) \\
&\leq \gamma \|Q_1 - Q_2\| + \frac{\gamma}{\beta} \log \left( 1 + \frac{\sum_{s'} P_2(s'|s, a) e^{-\beta V_1(s')} - \sum_{s'} P_1(s'|s, a) e^{-\beta V_1(s')}}{\sum_{s'} P_1(s'|s, a) e^{-\beta V_1(s')}} \right) \\
&\leq \gamma \|Q_1 - Q_2\| + \frac{\gamma}{\beta} \frac{\sum_{s'} P_2(s'|s, a) e^{-\beta V_1(s')} - \sum_{s'} P_1(s'|s, a) e^{-\beta V_1(s')}}{\sum_{s'} P_1(s'|s, a) e^{-\beta V_1(s')}} \\
&\leq \gamma \|Q_1 - Q_2\| + \frac{\gamma}{\beta} \frac{\left| \sum_{s'} [P_2(s'|s, a) - P_1(s'|s, a)] e^{-\beta V_1(s')} \right|}{e^{-\beta/(1-\gamma)}}.
\end{aligned}
$$

Rearranging the terms yields the asserted result:

$$
\|Q_1 - Q_2\| \leq \frac{\gamma}{1 - \gamma} \frac{e^{\beta/(1-\gamma)}}{\beta} \left| \sum_{s'} [P_2(s'|s, a) - P_1(s'|s, a)] e^{-\beta V_1(s')} \right|.
$$

**Case 2: $\beta > 0$ and $Q_1(s, a) < Q_2(s, a)$.** The proof is very similar to Case 1, but the extension $V_2(s) = V_1(s) + V_2(s) - V_1(s)$ is now done in the denominator instead:

$$
\begin{aligned}
\|Q_1 - Q_2\| &= \frac{\gamma}{\beta} \log \left( \frac{\sum_{s'} P_1(s'|s, a) e^{-\beta V_1(s')}}{\sum_{s'} P_2(s'|s, a) e^{-\beta V_2(s')}} \right) \\
&= \frac{\gamma}{\beta} \log \left( \frac{\sum_{s'} P_1(s'|s, a) e^{-\beta V_1(s')}}{\sum_{s'} P_2(s'|s, a) e^{-\beta V_1(s') - \beta(V_2(s') - V_1(s'))}} \right) \\
&\leq \frac{\gamma}{\beta} \log \left( e^{\beta \|V_1 - V_2\|} \frac{\sum_{s'} P_1(s'|s, a) e^{-\beta V_1(s')}}{\sum_{s'} P_2(s'|s, a) e^{-\beta V_1(s')}} \right) \\
&= \gamma \|V_1 - V_2\| + \frac{\gamma}{\beta} \log \left( \frac{\sum_{s'} P_1(s'|s, a) e^{-\beta V_1(s')}}{\sum_{s'} P_2(s'|s, a) e^{-\beta V_1(s')}} \right) \\
&\leq \gamma \|Q_1 - Q_2\| + \frac{\gamma}{\beta} \log \left( 1 + \frac{\sum_{s'} P_1(s'|s, a) e^{-\beta V_1(s')} - \sum_{s'} P_2(s'|s, a) e^{-\beta V_1(s')}}{\sum_{s'} P_2(s'|s, a) e^{-\beta V_1(s')}} \right) \\
&\leq \gamma \|Q_1 - Q_2\| + \frac{\gamma}{\beta} \frac{\sum_{s'} P_1(s'|s, a) e^{-\beta V_1(s')} - \sum_{s'} P_2(s'|s, a) e^{-\beta V_1(s')}}{\sum_{s'} P_2(s'|s, a) e^{-\beta V_1(s')}} \\
&\leq \gamma \|Q_1 - Q_2\| + \frac{\gamma}{\beta} \frac{\left| \sum_{s'} [P_1(s'|s, a) - P_2(s'|s, a)] e^{-\beta V_1(s')} \right|}{e^{-\beta/(1-\gamma)}},
\end{aligned}
$$

which again yields

$$
\|Q_1 - Q_2\| \leq \frac{\gamma}{1 - \gamma} \frac{e^{\beta/(1-\gamma)}}{\beta} \left| \sum_{s'} [P_2(s'|s, a) - P_1(s'|s, a)] e^{-\beta V_1(s')} \right|.
$$

**Case 3: $\beta < 0$ and $Q_1(s,a) > Q_2(s,a)$.** We have

$$
\begin{aligned}
\|Q_1 - Q_2\| &= \frac{\gamma}{|\beta|} \log \left( \frac{\sum_{s'} P_1(s'|s,a)e^{|\beta|V_1(s')}}{\sum_{s'} P_2(s'|s,a)e^{|\beta|V_2(s')}} \right) \\
&= \frac{\gamma}{|\beta|} \log \left( \frac{\sum_{s'} P_1(s'|s,a)e^{|\beta|V_1(s')}}{\sum_{s'} P_2(s'|s,a)e^{|\beta|V_1(s') - |\beta|(V_1(s') - V_2(s'))}} \right) \\
&\leq \frac{\gamma}{|\beta|} \log \left( \frac{\sum_{s'} P_1(s'|s,a)e^{|\beta|V_1(s')}}{\sum_{s'} P_2(s'|s,a)e^{|\beta|V_1(s') - |\beta|\|V_1 - V_2\|}} \right) \\
&= \gamma\|V_1 - V_2\| + \frac{\gamma}{|\beta|} \log \left( \frac{\sum_{s'} P_1(s'|s,a)e^{|\beta|V_1(s')}}{\sum_{s'} P_2(s'|s,a)e^{|\beta|V_1(s')}} \right) \\
&\leq \gamma\|Q_1 - Q_2\| + \frac{\gamma}{|\beta|} \log \left( 1 + \frac{\sum_{s'} P_1(s'|s,a)e^{|\beta|V_1(s')} - \sum_{s'} P_2(s'|s,a)e^{|\beta|V_1(s')}}{\sum_{s'} P_2(s'|s,a)e^{|\beta|V_1(s')}} \right) \\
&\leq \gamma\|Q_1 - Q_2\| + \frac{\gamma}{|\beta|} \Big| \sum_{s'} [P_1(s'|s,a) - P_2(s'|s,a)]e^{|\beta|V_1(s')} \Big| \\
&\leq \gamma\|Q_1 - Q_2\| + \frac{\gamma}{|\beta|} e^{|\beta|/(1-\gamma)} \Big| \sum_{s'} [P_2(s'|s,a) - P_1(s'|s,a)]e^{|\beta|[V_1(s') - \frac{1}{1-\gamma}]} \Big|,
\end{aligned}
$$

which implies

$$
\|Q_1 - Q_2\| \leq \frac{\gamma}{1-\gamma} \frac{e^{|\beta|/(1-\gamma)}}{|\beta|} \Big| \sum_{s'} [P_2(s'|s,a) - P_1(s'|s,a)]e^{|\beta|[V_1(s') - \frac{1}{1-\gamma}]} \Big|.
$$

**Case 4: $\beta < 0$ and $Q_2(s,a) > Q_1(s,a)$.** The proof of this case is similar to the other three cases and is omitted. $\qquad\square$

### D.4 Proof of Lemma 4

Let $N$ denote the number of calls to the generative model on each state-action pair such that the total number of calls is $SAN$. Let $\widehat{P}(s'|s,a)$ denote the plug-in estimator obtained from $N$ samples of $s' \sim P_{s,a}$, that is $\widehat{P}(s'|s,a) = \frac{1}{N} \sum_{n=1}^{N} \mathbb{1}_{\{X_n = s'\}}$, where $X_n$ taking values in $\mathcal{S}$ according to $P_{s,a}$.

**Lemma 4.** *Let $\pi$ be any fixed policy and $\tau > 0$. If $N > \frac{1}{2\tau^2}\left(1 - e^{-|\beta|/(1-\gamma)}\right)^2 \log(2SA/\delta)$, then it holds that*

$$
(i) \qquad \max_{s,a} \Big| \sum_{s'} [P_{s,a}(s') - \widehat{P}_{s,a}(s')]e^{-\beta V^\pi(s')} \Big| < \tau, \quad \text{with probability} \geq 1 - \delta, \quad \beta > 0;
$$

$$
(ii) \qquad \max_{s,a} \Big| \sum_{s'} [P_{s,a}(s') - \widehat{P}_{s,a}(s')]e^{-|\beta|(V^\pi(s') - \frac{1}{1-\gamma})} \Big| < \tau \quad \text{with probability} \geq 1 - \delta, \quad \beta < 0.
$$

*Proof.* We only prove the first claim —i.e., the case of $\beta > 0$— as the other case is proven using completely similar lines. We note that for the random variable $\sum_{s'} \mathbb{1}_{\{X_n = s'\}} e^{-\beta V^\pi(s')}$, we have that

$$
\mathbb{E}\left[ \sum_{s'} \mathbb{1}_{\{X_n = s'\}} e^{-\beta V^\pi(s')} \right] = \sum_{s'} \mathbb{E}\left[ \mathbb{1}_{\{X_n = s'\}} \right] e^{-\beta V^\pi(s')}
$$

$$
= \sum_{s'} P(s'|s,a) e^{-\beta V^\pi(s')}
$$

and that it is bounded in $[e^{-\beta/(1-\gamma)}, 1]$. Also, since

$$
\sum_{s'} \widehat{P}(s'|s,a) e^{-\beta V^\pi(s')} = \frac{1}{N} \sum_{n=1}^{N} \sum_{s'} \mathbb{1}_{\{X_n = s'\}} e^{-\beta V^\pi(s')},
$$

it follows directly from Hoeffding's inequality that

$$\mathbb{P}\left(\left|\sum_{s'}[P(s'|s,a) - \widehat{P}(s'|s,a)]e^{-\beta V^\pi(s')}\right| \geq \varepsilon\right) \leq 2\exp\left(-\frac{2N\varepsilon^2}{\left(1 - e^{-\beta/(1-\gamma)}\right)^2}\right).$$

Thus, by picking $N = \frac{1}{2\varepsilon^2}\left(1 - e^{-\beta/(1-\gamma)}\right)^2\log(2SA/\delta)$ and a union bound,

$$\mathbb{P}\left(\exists s,a : \left|\sum_{s'}[P(s'|s,a) - \widehat{P}(s'|s,a)]e^{-\beta V^\pi(s')}\right| \geq \varepsilon\right) \leq \delta.$$

$\square$

## E    Proof of Lemma 5

Next we prove a result that bounds the quality of a greedy policy with respect to the quality of the value-function for which the policy is greedy. The result is a generalization of Singh and Yee (1994) from the risk-neutral case to that of ERM, and the derivation follows the same lines. Throughout, we use the notation $\rho_{s,a}(V(s'))$ as shorthand notation for $\rho$ applied to the categorical random variable $X$ with support $\{V(s')\}_{s'\in\mathcal{S}}$ where $\mathbb{P}(X = V(s')) = P(s'|s,a)$.

**Lemma 5.** *Let $\alpha > 0$. Let $\overline{V} \in \mathbb{R}^S$ be a value function obeying $\|V^* - \overline{V}\| < \alpha$, and $\pi^G := \operatorname{argmax}_a[R(s,a) + \gamma\rho_{s,a}(\overline{V}(s'))]$ be the greedy policy with respect to $\overline{V}$. Then, $\|V^* - V^{\pi_G}\| \leq \frac{2\gamma}{1-\gamma}\alpha$.*

*Proof.* Let $\bar{s}$ be any state such that $\|V^* - V^G\| = V^*(\bar{s}) - V^G(\bar{s})$, where $V^G := V^{\pi_G}$. We then consider the two actions $a^* := \pi^*(\bar{s})$ and $a^G := \pi^G(\bar{s})$; ties can be broken arbitrarily. Since $\pi^G$ is greedy with respect to $\overline{V}$, we have that

$$R(\bar{s},a^*) + \gamma\rho_{\bar{s},a^*}(\overline{V}(s')) \leq R(\bar{s},a^G) + \gamma\rho_{\bar{s},a^G}(\overline{V}(s')).$$

By assumption, it holds for any $s \in \mathcal{S}$ that

$$V^*(s) - \alpha \leq \overline{V}(s) \leq V^*(s) + \alpha.$$

By monotonicity and translation invariance of $\rho$, we thus get

$$R(\bar{s},a^*) + \gamma\rho_{\bar{s},a^*}(\overline{V}(s')) \geq R(\bar{s},a^*) + \gamma\rho_{\bar{s},a^*}(V^*(s') - \alpha)$$
$$= R(\bar{s},a^*) + \gamma\rho_{\bar{s},a^*}(V^*(s')) - \gamma\alpha,$$

and similarly we have

$$R(\bar{s},a^G) + \gamma\rho_{\bar{s},a^G}(\overline{V}(s')) \leq R(\bar{s},a^G) + \gamma\rho_{\bar{s},a^G}(V^*(s')) + \gamma\alpha,$$

which collectively imply

$$R(\bar{s},a^*) - R(\bar{s},a^G) \leq 2\gamma\alpha + \gamma\left(\rho_{\bar{s},a^G}(V^*(s')) - \rho_{\bar{s},a^*}(V^*(s'))\right).$$

Finally, we obtain

$$V^*(\bar{s}) - V^G(\bar{s}) = R(\bar{s},a^*) - R(\bar{s},a^G) + \gamma\rho_{\bar{s},a^*}(V^*(s')) - \gamma\rho_{\bar{s},a^G}(V^G(s'))$$
$$\leq 2\gamma\alpha + \gamma\rho_{\bar{s},a^G}(V^*(s')) - \gamma\rho_{\bar{s},a^*}(V^*(s')) + \gamma\rho_{\bar{s},a^*}(V^*(s')) - \gamma\rho_{\bar{s},a^G}(V^G(s'))$$
$$= 2\gamma\alpha + \gamma\left(\rho_{\bar{s},a^G}(V^*(s')) - \rho_{\bar{s},a^G}(V^G(s'))\right)$$
$$= 2\gamma\alpha + \gamma\|V^* - V^G\|,$$

from which the result follows. $\square$

# F   Proofs of Lower Bounds

## F.1   Lower Bound on Bernoulli Likelihood Ratio

We revisit and develop a technical result that bounds the likelihood ratio of two samples under different hypotheses on a high probability event. Parts of the proof closely resemble parts of Lemma 17 in (Gheshlaghi Azar et al., 2013); however, we stress that our treatment fixes an error in the proof in (Gheshlaghi Azar et al., 2013), which however requires slightly stronger assumptions than those imposed in (Gheshlaghi Azar et al., 2013). In addition, while the result in (Gheshlaghi Azar et al., 2013) only considers $p \geq \frac{1}{2}$, ours deal with both cases of $p \geq \frac{1}{2}$ and $p < \frac{1}{2}$.

Let $p \in (0,1)$ and $\tilde{p} = \max\{p, 1-p\}$, and consider $\alpha \in (0, \frac{1-\tilde{p}}{5}]$. Consider two coins (Bernoulli random variables), one with bias $q = p$ and the other with bias $q = p + \alpha$. We consider two statistical hypotheses $H_0 : q = p$ and $H_1 : q = p + \alpha$.

Let $W$ be the outcome of flipping one of the coins $t$ times, and define the associated likelihood function under hypothesis $H_m$ with $m \in \{0,1\}$ as $L_m(w) := \mathbb{P}_m(W = w)$, where $\mathbb{P}_m(W = w)$, for every possible history of outcomes $w$, denotes the probability of observing $w$ under $H_m$. The likelihood function defines a random variable $L_m(W)$, where $W$ is the stochastic process of realized coin tosses.

Let $t \in \mathbb{N}$ and $\theta = \exp\left(-\frac{c_1 \alpha^2 t}{p(1-p)}\right)$. Let $k$ be the number of successes in the $t$ trials and

$$
\tilde{k} = \begin{cases} k & \text{if } p \geq \frac{1}{2} \\ t - k & \text{if } p < \frac{1}{2} . \end{cases}
$$

Finally, we define the event $\mathcal{E}$ as

$$
\mathcal{E} = \left\{ \tilde{p}t - \tilde{k} \leq \sqrt{2p(1-p)t \log\left(\frac{c_2}{2\theta}\right)} \right\},
$$

where $c_2 \geq 2$ is an arbitrary constant.

**Theorem 6.** *For $c_1 = 32$, it holds that $\frac{L_1(W)}{L_0(W)} \mathbb{1}_{\mathcal{E}} \geq \frac{2\theta}{c_2} \mathbb{1}_{\mathcal{E}}$ .*

*Proof.* We distinguish two cases depending on the value of $p$.

**Case 1:** $p \geq \frac{1}{2}$.   The likelihood ratio can be written as

$$
\frac{L_1(W)}{L_0(W)} = \frac{(p+\alpha)^k(1-p-\alpha)^{t-k}}{p^k(1-p)^{t-k}} = \left(1 + \frac{\alpha}{p}\right)^k \left(1 - \frac{\alpha}{1-p}\right)^{t-k}
$$
$$
= \left(1 + \frac{\alpha}{p}\right)^k \left(1 - \frac{\alpha}{1-p}\right)^{k\frac{1-p}{p}} \left(1 - \frac{\alpha}{1-p}\right)^{t - \frac{k}{p}} .
$$

We start by bounding the second factor using that $\log(1-x) \geq -x - x^2 + x^3$ for $x \in [0, \frac{1}{5}]$ (Lemma 7) and that $\exp(x) \geq 1 + x$ for all $x$ along with our assumption that $\alpha \leq \frac{1-p}{5}$:

$$
\left(1 - \frac{\alpha}{1-p}\right)^{\frac{1-p}{p}} \geq \exp\left(\frac{1-p}{p}\left[-\frac{\alpha}{1-p} - \frac{\alpha^2}{(1-p)^2} + \frac{\alpha^3}{(1-p)^3}\right]\right)
$$

$$
\geq 1 - \frac{1-p}{p}\left[\frac{\alpha}{1-p} + \frac{\alpha^2}{(1-p)^2} - \frac{\alpha^3}{(1-p)^3}\right]
$$

$$
= 1 - \frac{\alpha}{p} - \frac{\alpha^2}{p(1-p)} + \frac{\alpha^3}{p(1-p)^2}
$$

$$
\geq 1 - \frac{\alpha}{p} - \frac{\alpha^2}{p(1-p)} + \frac{\alpha^3}{p^2(1-p)}
$$

$$
= \left(1 - \frac{\alpha}{p}\right)\left(1 - \frac{\alpha^2}{p(1-p)}\right),
$$

where we have used that $p \geq 1 - p$.

Using this along with the fact that $k \leq t$ and $p \geq 1 - p$, it follows that

$$
\frac{L_1(W)}{L_0(W)} \geq \left(1 - \frac{\alpha^2}{p^2}\right)^k \left(1 - \frac{\alpha^2}{p(1-p)}\right)^k \left(1 - \frac{\alpha}{1-p}\right)^{t - \frac{k}{p}}
$$

$$
\geq \left(1 - \frac{\alpha^2}{p(1-p)}\right)^{2k} \left(1 - \frac{\alpha}{1-p}\right)^{t - \frac{k}{p}}
$$

$$
\geq \left(1 - \frac{\alpha^2}{p(1-p)}\right)^{2t} \left(1 - \frac{\alpha}{1-p}\right)^{t - \frac{k}{p}}.
$$

Note that we have $\alpha^2 \leq \frac{(1-p)^2}{25} \leq \frac{p(1-p)}{25} \leq \frac{p(1-p)}{2}$. Using this and the fact that $\log(1-x) \geq -2x$ for $x \in [0, \frac{1}{2}]$, we obtain

$$
\left(1 - \frac{\alpha^2}{p(1-p)}\right)^{2t} \geq \exp\left(-4t\frac{\alpha^2}{p(1-p)}\right) = \theta^{4/c_1} \geq \left(\frac{2\theta}{c_2}\right)^{4/c_1},
$$

where we have used that $c_2 \geq 2$.

Now on the event $\mathcal{E}$, we have that $t - \frac{k}{p} \leq \sqrt{2\frac{1-p}{p}t\log(\frac{c_2}{2\theta})}$. Using this along with the fact that $\frac{1}{c_1}\log(\frac{c_2}{2\theta}) \geq \frac{\alpha^2 t}{p(1-p)}$, which follows since

$$
\log\left(\frac{c_2}{2\theta}\right) = \log\left(\frac{c_2}{2}\exp\left[\frac{c_1\alpha^2 t}{p(1-p)}\right]\right) \geq \log\left(\exp\left[\frac{c_1\alpha^2 t}{p(1-p)}\right]\right) = \frac{c_1\alpha^2 t}{p(1-p)},
$$

we obtain that

$$
\left(1 - \frac{\alpha}{1-p}\right)^{t - \frac{k}{p}} \geq \left(1 - \frac{\alpha}{1-p}\right)^{\sqrt{2\frac{1-p}{p}t\log(\frac{c_2}{2\theta})}}
$$

$$
\geq \exp\left(-\frac{2\alpha}{1-p}\sqrt{2\frac{1-p}{p}t\log\left(\frac{c_2}{2\theta}\right)}\right)
$$

$$
= \exp\left(-\sqrt{\frac{8\alpha^2 t}{p(1-p)}\log\left(\frac{c_2}{2\theta}\right)}\right)
$$

$$
\geq \exp\left(-\sqrt{\frac{8}{c_1}\log\left(\frac{c_2}{2\theta}\right)}\right)
$$

$$
= \left(\frac{2\theta}{c_2}\right)^{\sqrt{8/c_1}}.
$$

Putting these together, we observe that

$$\frac{L_1(W)}{L_0(W)}\mathbb{1}_\mathcal{E} \geq \left(\frac{2\theta}{c_2}\right)^{\sqrt{8/c_1}+4/c_1}\mathbb{1}_\mathcal{E},$$

so that choosing $c_1 = 32$ yields the claimed result:

$$\frac{L_1(W)}{L_0(W)}\mathbb{1}_\mathcal{E} \geq \frac{2\theta}{c_2}\mathbb{1}_\mathcal{E}.$$

**Case 2: $p < \frac{1}{2}$.**  Define $m = t - k$, which corresponds to the number of failed coin flips. Hence,

$$\frac{L_1(W)}{L_0(W)} = \frac{(1-p-\alpha)^m(p+\alpha)^{t-m}}{(1-p)^m p^{t-m}} = \left(1-\frac{\alpha}{1-p}\right)^m\left(1+\frac{\alpha}{p}\right)^{t-m}$$

$$= \left(1-\frac{\alpha}{1-p}\right)^m\left(1+\frac{\alpha}{p}\right)^{\frac{mp}{1-p}}\left(1+\frac{\alpha}{p}\right)^{t-\frac{m}{1-p}}.$$

Again, using $\exp(x) \geq x+1$ for all $x \in \mathbb{R}$ and using that $\log(1+x) \geq x - x^2$ for all $x \geq 0$, we get that

$$\left(1+\frac{\alpha}{p}\right)^{\frac{p}{1-p}} \geq \exp\left(\frac{p}{1-p}\left[\frac{\alpha}{p}-\frac{\alpha^2}{p^2}\right]\right)$$

$$\geq 1 + \frac{\alpha}{1-p} - \frac{\alpha^2}{p(1-p)}$$

$$\geq 1 + \frac{\alpha}{1-p} - \frac{\alpha^2}{p(1-p)} - \frac{\alpha^3}{p(1-p)^2}$$

$$= \left(1+\frac{\alpha}{1-p}\right)\left(1-\frac{\alpha^2}{p(1-p)}\right).$$

Using this along with the fact that $1-p > p$ and $m \leq t$, we have

$$\frac{L_1(W)}{L_0(W)} \geq \left(1-\frac{\alpha^2}{(1-p)^2}\right)^m\left(1-\frac{\alpha^2}{p(1-p)}\right)^m\left(1+\frac{\alpha}{p}\right)^{t-\frac{m}{1-p}}$$

$$\geq \left(1-\frac{\alpha^2}{p(1-p)}\right)^{2t}\left(1-\frac{\alpha}{p}\right)^{t-\frac{m}{1-p}}.$$

Again, using $\log(1-x) \geq -2x$ for $x \in [0,\frac{1}{2}]$, we get that

$$\left(1-\frac{\alpha^2}{p(1-p)}\right)^{2t} \geq \exp\left(-4t\frac{\alpha^2}{p(1-p)}\right) \geq \theta^{4/c_1} \geq \left(\frac{2\theta}{c_2}\right)^{4/c_1}.$$

On the event $\mathcal{E}$, we have that $t-\frac{m}{1-p} \leq \sqrt{\frac{2pt}{1-p}\log(\frac{c_2}{2\theta})}$. Using this along with the fact that $\frac{1}{c_1}\log(\frac{c_2}{2\theta}) \geq \frac{\alpha^2 t}{p(1-p)}$, we get on the event $\mathcal{E}$ that

$$\left(1-\frac{\alpha}{p}\right)^{t-\frac{m}{1-p}} \geq \left(1-\frac{\alpha}{p}\right)^{\sqrt{\frac{2p}{1-p}t\log(\frac{c_2}{2\theta})}}$$

$$\geq \exp\left(-2\sqrt{\frac{2t\alpha^2}{p(1-p)}\log\left(\frac{c_2}{2\theta}\right)}\right)$$

$$\geq \exp\left(-\sqrt{\frac{8}{c_1}\log\left(\frac{c_2}{2\theta}\right)}\right)$$

$$= \left(\frac{2\theta}{c_2}\right)^{\sqrt{8/c_1}}.$$

We thus get the desired result for $c_1 = 32$:

$$\frac{L_1(W)}{L_0(W)}\mathbb{1}_{\mathcal{E}} \geq \left(\frac{2\theta}{c_2}\right)^{\sqrt{8/c_1}+4/c_1}\mathbb{1}_{\mathcal{E}} \geq \frac{2\theta}{c_2}\mathbb{1}_{\mathcal{E}}.$$

$\square$

### F.2 Lower Bound for Q-value Learning: Proof of Theorem 3

The proof is adapted from the lower bound analysis of (Gheshlaghi Azar et al., 2013), originally developed for value learning in risk-neutral RL, with necessary modifications to accommodate recursive ERM. For completeness, we present the full proof.

For a lower bound we construct the following class of MDPs with $S' := S + 2$ states and $A$ actions, where the first states are labelled as $s_1, \ldots, s_S, s^G, s^B$, and the actions are labelled as $a_1, \ldots, a_A$. The states $s^G$ and $s^B$ are absorbing under any actions and $R(s^G, a) = 1$ for all $j$ and $R(s^B, a) = 0$ for all $a \in \mathcal{A}$. For the states $s \in \{s_1, \ldots, s_S\}$, we have that $R(s, a) = 0$ for all $a \in \mathcal{A}$. We have $SA$ state-action pair combinations from $\{s_1, \ldots, s_S\} \times \mathcal{A} =: Z$ on which we assume some ordering, allowing us to write $z_i, i \in [SA]$. Finally, for all state-action pairs $z_i \in [SA]$, we have $P(s^G|z_i) = q_i$ and $P(s^B|z_i) = 1 - q_i$ for some $q_i \in [0, 1]$. Denote the collection of all such MDPs by $\mathbb{M}$.

The structure of this class of MDPs allows us to get lower bounds on the samples needed to learn the $Q$-value of each state-action pair $z_i$ and then use the fact that samples used to learn the $Q$-values for different state-action pairs bring no information on each other to get the final bound.

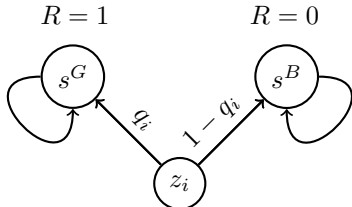

$R = 1 \qquad\qquad R = 0$

Figure 3: Dynamics and rewards of the hard-to-learn MDP class

For any state-action pair $z_i$, we can explicitly calculate the state-action value-functions

$$Q(z_i) = \frac{-\gamma}{\beta}\log\left(q_i e^{-\beta\frac{1}{1-\gamma}} + 1 - q_i\right),$$

and $Q(s^G, a) = \frac{1}{1-\gamma}$ and $Q(s^B, a) = 0$ for all $a$.

Fix any index $i \in [SA]$, and consider the two hypotheses $H_0^i : q_i = p$ and $H_1^i : q_i = p + \alpha$, where $p$ and $\alpha$ are given by

$$p = \begin{cases} 1 - e^{-\beta\frac{1}{1-\gamma}} & \text{for } \beta > 0, \\ e^{-|\beta|\frac{1}{1-\gamma}} & \text{for } \beta < 0, \end{cases}$$

and $\alpha = 8\varepsilon\frac{|\beta|}{\gamma}\left(e^{|\beta|\frac{1}{1-\gamma}} - 1\right)^{-1}$, for any $\varepsilon$ obeying $\varepsilon < \frac{1}{40}\frac{\gamma}{|\beta|}\left(1 - e^{-|\beta|\frac{1}{1-\gamma}}\right)$.

We use $M_0$ to denote an MDP where $H_0^i$ holds, and $M_1$ to an MDP where $H_1^i$ holds. Further, $\mathbb{E}_0$ and $\mathbb{P}_0$ denote, respectively, the expectation and probability operators under $H_0^i$, and $\mathbb{E}_1$ and $\mathbb{P}_1$ are similarly defined for $H_1^i$. Fix any $(\varepsilon, \delta)$-correct $Q$-algorithm $\mathcal{U}$. We start by showing that with these parameter we have that $Q_{M_1}^*(z_i) - Q_{M_0}^*(z_i) > 2\varepsilon$, which we do by casing on the sign of $\beta$:

**Case 1:** $\beta < 0$. In this case $p = e^{-|\beta|\frac{1}{1-\gamma}}$. We then have

$$
\begin{aligned}
Q^*_{M_1}(z_i) - Q^*_{M_0}(z_i) &= \frac{\gamma}{|\beta|} \log\left( \frac{(p+\alpha)e^{|\beta|\frac{1}{1-\gamma}} + 1 - p - \alpha}{pe^{|\beta|\frac{1}{1-\gamma}} + 1 - p} \right) \\
&= \frac{\gamma}{|\beta|} \log\left( 1 + \frac{\alpha(e^{|\beta|\frac{1}{1-\gamma}} - 1)}{pe^{|\beta|\frac{1}{1-\gamma}} + 1 - p} \right) \\
&\geq \frac{\gamma}{|\beta|} \frac{\alpha}{2} \frac{e^{|\beta|\frac{1}{1-\gamma}} - 1}{pe^{|\beta|\frac{1}{1-\gamma}} + 1 - p} \\
&> \frac{\gamma}{|\beta|} \frac{\alpha}{4} (e^{|\beta|\frac{1}{1-\gamma}} - 1) \\
&= 2\varepsilon \,,
\end{aligned}
$$

where we have used that $p = e^{-|\beta|\frac{1}{1-\gamma}}$ and the fact that $\log(1+x) \geq \frac{x}{2}$ for $x \in [0,1]$.

**Case 2:** $\beta > 0$. The case for $\beta > 0$ is similar, although in this case we have $p = 1 - e^{-\beta\frac{1}{1-\gamma}}$ and use the inequality $\log(1+x) \leq x$ for all $x > -1$ to get that

$$
\begin{aligned}
Q^*_{M_1}(z_i) - Q^*_{M_0}(z_i) &= -\frac{\gamma}{\beta} \log\left( \frac{(p+\alpha)e^{-\beta\frac{1}{1-\gamma}} + 1 - p - \alpha}{pe^{-\beta\frac{1}{1-\gamma}} + 1 - p} \right) \\
&= -\frac{\gamma}{\beta} \log\left( 1 - \frac{\alpha(1 - e^{-\beta\frac{1}{1-\gamma}})}{1 - p + pe^{-\beta\frac{1}{1-\gamma}}} \right) \\
&= -\frac{\gamma}{\beta} \log\left( 1 - \frac{\alpha(1 - e^{-\beta\frac{1}{1-\gamma}})}{(1-p)e^{-\beta\frac{1}{1-\gamma}}} \right) \\
&\geq \frac{\gamma}{\beta} \alpha \frac{1 - e^{-\beta\frac{1}{1-\gamma}}}{(1+p)e^{-\beta\frac{1}{1-\gamma}}} \\
&\geq \frac{\gamma}{\beta} \alpha \frac{1 - e^{-\beta\frac{1}{1-\gamma}}}{2e^{-\beta\frac{1}{1-\gamma}}} \\
&\geq \frac{\gamma}{\beta} \alpha \frac{e^{\beta\frac{1}{1-\gamma}} - 1}{2} \\
&= 4\varepsilon \,.
\end{aligned}
$$

In particular, this means that the events $B_0 := \{|Q^*_{M_0}(z_i) - Q^{\mathcal{U}}_t(z_i)| \leq \varepsilon\}$ and $B_1 := \{|Q^*_{M_1}(z_i) - Q^{\mathcal{U}}_t(z_i)| \leq \varepsilon\}$ are disjoint events. Let $t$ be the number of times the algorithm tries $z_i$. Since $\mathcal{U}$ is $(\varepsilon, \delta)$-correct, it holds that $\mathbb{P}_0(B_0) \geq 1 - \delta \geq \frac{3}{4}$.

Let $k$ be the number of transitions from $z_i$ to $s^G$ in the $t$ trials. We then define $\tilde{k}, \tilde{p}$ and $\theta$ as

$$
\theta := \exp\left( -\frac{32\alpha^2 t}{p(1-p)} \right), \qquad \tilde{p} = \max\{p, 1-p\}, \qquad \tilde{k} := \begin{cases} k & \text{if } p \geq \frac{1}{2} \\ t - k & \text{if } p < \frac{1}{2} \end{cases}
$$

and the event

$$
\mathcal{E} = \left\{ \tilde{p}t - \tilde{k} \leq \sqrt{2p(1-p)t \log\left( \frac{8}{2\theta} \right)} \right\},
$$

for which, we have $\mathbb{P}_0(\mathcal{E}) > \frac{3}{4}$ by Lemma 16 in (Gheshlaghi Azar et al., 2013) and thus $\mathbb{P}_0(B_0 \cap \mathcal{E}) > \frac{1}{2}$. Now by Theorem 6, we get that

$$
\mathbb{P}_1(B_0) \geq \mathbb{P}_1(B_0 \cap \mathcal{E}) = \mathbb{E}_1[\mathbb{1}_{\mathcal{E}} \mathbb{1}_{B_0}] = \mathbb{E}_0\left[ \frac{L_1}{L_0} \mathbb{1}_{\mathcal{E}} \mathbb{1}_{B_0} \right] \geq \frac{\theta}{4} \mathbb{E}_0[\mathbb{1}_{\mathcal{E}} \mathbb{1}_{B_0}] = \frac{\theta}{4} \mathbb{P}_0(\mathcal{E} \cap B_0) \geq \frac{\theta}{8} \,.
$$

Requiring $\frac{\theta}{8} > \delta$ and solving for $t$ yield

$$t < \frac{p(1-p)}{32\alpha^2} \log\left(\frac{1}{8\delta}\right).$$

Noting that

$$\frac{p(1-p)}{\alpha^2} = \frac{\gamma^2}{|\beta|^2} \frac{e^{-|\beta|\frac{1}{1-\gamma}}(1 - e^{-|\beta|\frac{1}{1-\gamma}})}{64\varepsilon^2}(e^{|\beta|\frac{1}{1-\gamma}} - 1)^2 \geq \frac{\gamma^2}{64\varepsilon^2} \frac{e^{|\beta|\frac{1}{1-\gamma}} - 3}{|\beta|^2},$$

we conclude that if the algorithm $\mathcal{U}$ tries the state-action pair $z_i$ less than

$$\widetilde{T}(\varepsilon,\delta) := \frac{\gamma^2}{2048\varepsilon^2} \frac{e^{|\beta|\frac{1}{1-\gamma}} - 3}{|\beta|^2} \log\left(\frac{1}{8\delta}\right)$$

times under the hypothesis $H_0^i$, then $\mathbb{P}_1(B_0) > \delta$ and $B_0 \subset B_1^c$.

Let $n := SA$. If the number of total samples is less than $\frac{n}{2}\widetilde{T}(\varepsilon,\delta)$, there must be at least $n/2$ state-action pairs $z_i$ that has been tried no more than $\widetilde{T}(\varepsilon,\delta)$ times, which without loss of generality are assumed to be the state-action pairs $\{z_i\}_{i=1}^{n/2}$.

Let $T_i$ be the number of times the algorithm has tried $z_i$ for $i \leq n/2$. Due to the structure of the MDPs in $\mathbb{M}$, it is sufficient to consider only the algorithms that output an estimate of $Q_{T_i}^{\mathcal{U}}$ based on samples from $z_i$, since any other samples can yield no information on $Q^*(z_i)$.

Hence, introducing the events $\Lambda_i := \{|Q_{M_1}^*(z_i) - Q_{T_i}^{\mathcal{U}}(z_i)| > \varepsilon\}$, we have that $\Lambda_i$ and $\Lambda_j$, for distinct $i, j \in [n/2]$, are conditionally independent given $T_i$ and $T_j$. We then have

$$\mathbb{P}_1\left(\{\Lambda_i^c\}_{1\leq i\leq n/2} \cap \{T_i \leq \widetilde{T}(\varepsilon,\delta)\}_{1\leq i\leq n/2}\right)$$

$$= \sum_{t_1=0}^{\widetilde{T}(\varepsilon,\delta)} \cdots \sum_{t_{n/2}=0}^{\widetilde{T}(\varepsilon,\delta)} \mathbb{P}_1(\{T_i = t_i\}_{1\leq i\leq n/2})\mathbb{P}_1(\{\Lambda_i^c\}_{1\leq i\leq n/2} \cap \{T_i = t_i\}_{1\leq i\leq n/2})$$

$$= \sum_{t_1=0}^{\widetilde{T}(\varepsilon,\delta)} \cdots \sum_{t_{n/2}=0}^{\widetilde{T}(\varepsilon,\delta)} \mathbb{P}_1(\{T_i = t_i\}_{1\leq i\leq n/2}) \prod_{1\leq i\leq n/2} \mathbb{P}_1(\Lambda_i^c \cap \{T_i = t_i\})$$

$$= \sum_{t_1=0}^{\widetilde{T}(\varepsilon,\delta)} \cdots \sum_{t_{n/2}=0}^{\widetilde{T}(\varepsilon,\delta)} \mathbb{P}_1(\{T_i = t_i\}_{1\leq i\leq n/2})(1-\delta)^{n/2},$$

where we have used the law of total probability in the first equality, and the second equality holds by independence. This result implies that

$$\mathbb{P}_1\left(\{\Lambda_i^c\}_{1\leq i\leq n/2} \,\middle|\, \{T_i \leq \widetilde{T}(\varepsilon,\delta)\}_{1\leq i\leq n/2}\right) \leq (1-\delta)^{\frac{n}{2}}.$$

Thus, if the total number $T$ of transitions is less than $\frac{n}{2}\widetilde{T}(\varepsilon,\delta)$, then

$$\mathbb{P}_1(\|Q^* - Q_T^{\mathcal{U}}\| > \varepsilon) \geq \mathbb{P}_1\left(\bigcup_{i\in[SA]} \Lambda_i\right)$$

$$= 1 - \mathbb{P}_1\left(\bigcap_{1\leq i\leq n/2} \Lambda_i^c\right)$$

$$\geq 1 - \mathbb{P}_1\left(\{\Lambda_i^c\}_{1\leq i\leq n/2} \,\middle|\, \{T_{z_i} \leq \widetilde{T}(\varepsilon,\delta)\}_{1\leq i\leq n/2}\right)$$

$$\geq 1 - (1-\delta)^{n/2}$$

$$\geq \frac{\delta n}{4},$$

when $\delta \frac{n}{2} \le 1$ by Lemma 8. By setting $\delta' = \delta \frac{n}{4}$ and substituting back $S'$ we obtain the result. This shows that if the number of samples is smaller than

$$T = \frac{(S'-2)A}{4096} \frac{\gamma^2}{\varepsilon^2} \frac{e^{|\beta|\frac{1}{1-\gamma}} - 3}{|\beta|^2} \log\left(\frac{(S'-2)A}{32\delta}\right) \tag{13}$$

on the MDP corresponding to the hypothesis $H_0 : \{H_0^i | 1 \le i \le n\}$, it holds that $\mathbb{P}_1(\|Q_{M_1}^* - Q_T^{\mathcal{U}}\| > \varepsilon) > \delta'$. This completes the proof. $\qquad\square$

### F.3  Lower Bound for Policy Learning: Proof of Theorem 4

The proof is adapted from the lower bound analysis of (Gheshlaghi Azar et al., 2013), originally developed for value learning in risk-neutral RL, with necessary modifications to accommodate policy learning under recursive ERM. For completeness, we present the full proof.

For a lower bound we construct the following class of MDPs with $S' := S + 2$ states and $A' := A + 1$ actions, where the first states are labelled as $s_1, \ldots, s_S, s^G, s^B$, and the actions are labelled as $a_0, a_1, \ldots, a_A$. The states $s^G$ and $s^B$ are absorbing under any actions and $R(s^G, a) = 1$ for all $j$ and $R(s^B, a) = 0$ for all $a \in \mathcal{A}$. For the states $s \in \{s_1, \ldots, s_S\}$, we have that $R(s, a) = 0$ for all $a \in \mathcal{A}$.

From the state $s_i$ with probabilities that depend on the action taken the agent will then end up in either a good state $s^G$ which is absorbing and yields the maximal unit reward under all actions or in the bad state $s^B$ which is also absorbing but which yields no reward under any action. The different MDPs thus differ only in their transition probabilities in the choice states $s_i$.

Fix an index $1 \le i \le S$. We then consider the following set of possible parameters called hypotheses $H_l^i, l \in \{0, 1, 2, \ldots, A\}$ given by

$$
\begin{aligned}
H_0^i &: q(s_i, a_0) = p + \alpha & q(s_i, a) = p \text{ for } a \ne a_0 \\
H_l &: q(s_i, a_0) = p + \alpha & q(s_i, a) = p \text{ for } a \notin \{a_0, l\} & q(s_i, a_l) = p + 2\alpha,
\end{aligned}
$$

where $p$ and $\alpha$ are given by

$$\alpha = \frac{5|\beta|}{\gamma} \frac{\varepsilon}{e^{|\beta|\frac{1}{1-\gamma}} - 1} \quad \text{and} \quad p = \begin{cases} 1 - e^{-\beta\frac{1}{1-\gamma}} & \beta > 0, \\ e^{-|\beta|\frac{1}{1-\gamma}} & \beta < 0, \end{cases}$$

where we allow for $0 < \varepsilon < \frac{\gamma}{50|\beta|}\left(1 - e^{-|\beta|\frac{1}{1-\gamma}}\right)$, which ensures that $\alpha \le \frac{e^{-|\beta|\frac{1}{1-\gamma}}}{10}$.

Consider a fixed hypothesis $H_l^i$ for some $l \ne 0$ and the sub-MDP that only consists of the states $\{s_i, s^G, s^B\}$. Here the optimal action is $a^* = a_l$, the second best action is $a_0$ and all other actions are even worse so the value-error over all states in the triplet for any suboptimal choice of actions will be at least as large as $V^*(s_i) - V^0(s_i)$ where $V^0$ is the value by choosing $a = 0$. We now show that any non-optimal action is $\varepsilon$-bad on $s_i$.

**Case 1:** $\beta > 0$. We have

$$
\begin{aligned}
V^*(s_i) - V^0(s_i) &= -\frac{\gamma}{\beta} \log \left( \frac{(p+2\alpha)e^{-\beta\frac{1}{1-\gamma}} + 1 - p - 2\alpha}{(p+\alpha)e^{-\beta\frac{1}{1-\gamma}} + 1 - p - \alpha} \right) \\
&= \frac{-\gamma}{\beta} \log \left( 1 - \alpha \frac{1 - e^{-\beta\frac{1}{1-\gamma}}}{pe^{-\beta\frac{1}{1-\gamma}} + 1 - p - \alpha(1 - e^{-\beta\frac{1}{1-\gamma}})} \right) \\
&> \frac{\gamma}{\beta} \alpha \frac{1 - e^{-\beta\frac{1}{1-\gamma}}}{pe^{-\beta\frac{1}{1-\gamma}} + 1 - p - \alpha(1 - e^{-\beta\frac{1}{1-\gamma}})} \\
&\geq \frac{\gamma}{\beta} \alpha \frac{1 - e^{-\beta\frac{1}{1-\gamma}}}{pe^{-\beta\frac{1}{1-\gamma}} + 1 - p} \\
&= \frac{\gamma}{\beta} \alpha \frac{1 - e^{-\beta\frac{1}{1-\gamma}}}{(1+p)e^{-\beta\frac{1}{1-\gamma}}} \\
&\geq \frac{\gamma}{\beta} \alpha \frac{1 - e^{-\beta\frac{1}{1-\gamma}}}{2e^{-\beta\frac{1}{1-\gamma}}} \\
&= \frac{\gamma}{2\beta} \alpha (1 - e^{-\beta\frac{1}{1-\gamma}}) \geq \varepsilon,
\end{aligned}
$$

where we have used $\log(1 + x) > x$ for $x \in (-1, \infty) \setminus \{0\}$.

**Case 2:** $\beta < 0$. We have

$$
\begin{aligned}
V^*(s_i) - V^0(s_i) &= \frac{\gamma}{|\beta|} \log \left( \frac{(p+2\alpha)e^{|\beta|\frac{1}{1-\gamma}} + 1 - p - 2\alpha}{(p+\alpha)e^{|\beta|\frac{1}{1-\gamma}} + 1 - p - \alpha} \right) \\
&= \frac{\gamma}{|\beta|} \log \left( 1 + \alpha \frac{e^{|\beta|\frac{1}{1-\gamma}} - 1}{pe^{-\beta\frac{1}{1-\gamma}} + 1 - p + \alpha(e^{|\beta|\frac{1}{1-\gamma}} - 1)} \right) \\
&> \frac{\gamma}{2|\beta|} \alpha \frac{e^{|\beta|\frac{1}{1-\gamma}} - 1}{pe^{-\beta\frac{1}{1-\gamma}} + 1 - p + \alpha(e^{|\beta|\frac{1}{1-\gamma}} - 1)} \\
&\geq \frac{\gamma}{2|\beta|} \alpha \frac{e^{|\beta|\frac{1}{1-\gamma}} - 1}{2 + \alpha(e^{|\beta|\frac{1}{1-\gamma}} - 1)} \\
&\geq \frac{\gamma}{2|\beta|} \alpha \frac{e^{|\beta|\frac{1}{1-\gamma}} - 1}{2 + \frac{1}{10}} \\
&= \frac{5}{21} \frac{\gamma}{|\beta|} \alpha (e^{|\beta|\frac{1}{1-\gamma}} - 1) \geq \varepsilon,
\end{aligned}
$$

where we have used $\log(1 + x) > \frac{x}{2}$ for $x \in (0, 1)$.

Now having shown that all non-optimal actions are $\varepsilon$-bad, we wish to show that any algorithm that is $(\varepsilon, \delta)$-correct on $H_0^i$, i.e. choosing the action $a_0$ with probability at least $1 - \delta$, will also have a probability of choosing $a_0$ on $H_l^i$ that is larger than $\delta$ provided that $a_l$ is not tried sufficiently many times under $H_0^i$.

Let $\mathbb{P}_l$ and $\mathbb{E}_l$ denote the probability operator and expectation operator under the hypothesis $H_l^i$. Let $t := t_l^i$ be the number of times the algorithm tries action $l$ in $s_i$ under $H_0$. Assuming that $\delta \in (0, \frac{1}{4})$ and using that the algorithm is $(\varepsilon, \delta)$-correct we have that $\mathbb{P}_0(B) \geq 1 - \delta \geq \frac{3}{4}$ where $B = \{\pi^\mathcal{U}(s_i) = a_0\}$ is the event that the algorithm outputs the action $a_0$.

Let $\theta = \exp\left(-\frac{32\alpha^2 t}{p(1-p)}\right)$. Fix some $t \in \mathbb{N}$ and let $k$ be the number of transitions to $s_i^G$ in the $t$ trials and

$$
\tilde{k} = \begin{cases} k & \text{if } p \geq \frac{1}{2} \\ t - k & \text{if } p < \frac{1}{2}. \end{cases}
$$

Finally, we define the event $\mathcal{E}$ as

$$\mathcal{E} = \left\{ \tilde{p}t - \tilde{k} \leq \sqrt{2p(1-p)t \log\left(\frac{8}{2\theta}\right)} \right\}. \tag{14}$$

From the Chernoff-Hoeffding bound and as shown in (Gheshlaghi Azar et al., 2013), we have that $\mathbb{P}_0(\mathcal{E}) > \frac{3}{4}$, and thus, $\mathbb{P}_0(B \cap \mathcal{E}) > \frac{1}{2}$. From Theorem 6, we get that

$$\mathbb{P}_1(B) \geq \mathbb{P}_1(B \cap \mathcal{E}) = \mathbb{E}_1[\mathbb{1}_B \mathbb{1}_{\mathcal{E}}] \geq \mathbb{E}_0\left[\frac{L_1(W)}{L_0(W)}\mathbb{1}_{\mathcal{E}}\mathbb{1}_B\right] \geq \mathbb{E}_0\left[\frac{\theta}{4}\mathbb{1}_{\mathcal{E}}\mathbb{1}_B\right] = \frac{\theta}{4}\mathbb{P}_0(\mathcal{E} \cap B) \geq \frac{\theta}{8}. \tag{15}$$

Now solving for $\frac{\theta}{8} > \delta$, we see that if

$$t < \widetilde{T}(\varepsilon, \delta) := \frac{1}{800}\log\left(\frac{1}{8\delta}\right)\frac{\gamma^2}{\varepsilon^2} \cdot \frac{e^{|\beta|\frac{1}{1-\gamma}} - 3}{|\beta|^2} \tag{16}$$

then $\mathbb{P}_1(B) > \delta$ and the event $B$ is containing the event that the algorithm does not choose the optimal action $a_l$.

Since this holds for all the $A$ hypotheses $H_l^i, l = 1, 2, \ldots, A$, it follows that the algorithm needs at least $\widetilde{T}(\varepsilon, \delta) := A\widetilde{T}(\varepsilon, \delta)$ samples to be $(\varepsilon, \delta)$-correct on the state $s_i$.

Next we use the fact that the structure of the MDPs is such that the information used to determine $\pi^*(s_i)$ carries no information to determine $\pi^*(s_j)$ for $i \neq j$.

If the number of total transition samples is less than $\frac{S}{2}\widetilde{T}(\varepsilon, \delta)$, then there must be at least $\frac{S}{2}$ states in the set $\{s_i\}_{i=1}^S$ for which some action (apart from $a_0$) has been tried no more than $\widetilde{T}(\varepsilon, \delta)$ times. Without loss of generality, we might assume that these are the states $\{s_i\}_{i=1}^{S/2}$ and that it is action $a_1$ that has been tried out at most $\widetilde{T}(\varepsilon, \delta)$ times in each of these states.

Let $T_i$ be the number of times the algorithm has tried sampled any action on $s_i$ for $i \leq S/2$ Due to the structure of the MDPs in $\mathbb{M}$ it is sufficient to consider only the algorithms that yields an estimate of $\pi_{T_i}^{\mathcal{U}}$ based on samples from $s_i$ since any other samples can yield no information on $\pi^*(s_i)$.

Let us define the events $\Lambda_i := \{|V_{M_1}^*(s_i) - V^{\pi_{T_i}^{\mathcal{U}}}(s_i)| > \varepsilon\}$ for $i = 1, \ldots, S$. Then, we have that $\Lambda_i$ and $\Lambda_j$ are conditionally independent given $T_i$ and $T_j$. We then have that for the MDP $M_1 \in \mathbb{M}$ –the one corresponding to the hypothesis $H_1 := \{H_1^i | 1 \leq i \leq n\}$– it holds that

$$\mathbb{P}\left(\{\Lambda_i^c\}_{1 \leq i \leq S/2} \cap \{T_i \leq \widetilde{T}(\varepsilon, \delta)\}_{1 \leq i \leq S/2}\right) = \sum_{t_1=0}^{\widetilde{T}(\varepsilon,\delta)} \cdots \sum_{t_{S/2}=0}^{\widetilde{T}(\varepsilon,\delta)} \mathbb{P}\left(\{T_i = t_i\}_{1 \leq i \leq S/2}\right)\mathbb{P}\left(\{\Lambda_i^c\}_{1 \leq i \leq S/2} \cap \{T_i = t_i\}_{1 \leq i \leq S/2}\right)$$

$$= \sum_{t_1=0}^{\widetilde{T}(\varepsilon,\delta)} \cdots \sum_{t_{S/2}=0}^{\widetilde{T}(\varepsilon,\delta)} \mathbb{P}\left(\{T_i = t_i\}_{1 \leq i \leq S/2}\right) \prod_{1 \leq i \leq S/2} \mathbb{P}\left(\Lambda_i^c \cap \{T_i = t_i\}\right)$$

$$= \sum_{t_1=0}^{\widetilde{T}(\varepsilon,\delta)} \cdots \sum_{t_{S/2}=0}^{\widetilde{T}(\varepsilon,\delta)} \mathbb{P}\left(\{T_i = t_i\}_{1 \leq i \leq S/2}\right)(1 - \delta)^{S/2},$$

where the first line follows from the law of total probability, and the second line from independence. We now have directly that

$$\mathbb{P}\left(\{\Lambda_i^c\}_{1 \leq i \leq S/2}\Big|\{T_i \leq \widetilde{T}(\varepsilon, \delta)\}_{1 \leq i \leq S/2}\right) \leq (1 - \delta)^{\frac{S}{2}}.$$

Thus, if the total number of transitions $T$ is less than $\frac{S}{2}\widetilde{T}(\varepsilon, \delta)$ on the MDP $M_0$ corresponding to the hypothesis $H_0 : \{H_0^i | 1 \le i \le n\}$, then on $M_1$ it holds that

$$
\begin{aligned}
\mathbb{P}(\|V^* - V^{\pi_T^{\mathcal{U}}}\| > \varepsilon) &\ge \mathbb{P}\left( \bigcup_{1 \le i \le S/2} \Lambda_i \right) \\
&= 1 - \mathbb{P}\left( \bigcap_{1 \le i \le S/2} \Lambda_i^c \right) \\
&\ge 1 - \mathbb{P}\left( \{\Lambda_i^c\}_{1 \le i \le S/2} \,\Big|\, \{T_{z_i} \le \widetilde{T}(\varepsilon, \delta)\}_{1 \le i \le S/2} \right) \\
&\ge 1 - (1 - \delta)^{S/2} \\
&\ge \frac{\delta S}{4},
\end{aligned}
$$

when $\frac{\delta S}{2} \le 1$ by Lemma 8. By setting $\delta' = \delta \frac{S}{4}$ and substituting back $S'$ and $A'$, we obtain the result. This shows that if the number of samples is smaller than

$$
T = \frac{(S' - 2)(A' - 1)}{1600} \log\left( \frac{S' - 2}{32\delta} \right) \frac{\gamma^2}{\varepsilon^2} \cdot \frac{e^{|\beta|^{\frac{1}{1-\gamma}}} - 3}{|\beta|^2}
$$

on $M_0$, then on $M_1$ it holds that $\mathbb{P}(\|V^* - V^{\pi_T^{\mathcal{U}}}\| > \varepsilon) > \delta$. This completes the proof. $\qquad\square$

