# OpenReview forum: "Recursive Entropic Risk Optimization in Discounted MDPs: Sample Complexity Bounds with a Generative Model"
_TMLR — Accepted by TMLR_

### Review · Reviewer_nMh1 · 2026-03-04

**Summary Of Contributions:**

The authors propose a novel model-based value iteration algorithm to optimize the entropic risk measure (ERM). The authors derive PAC-type bounds of the sample complexity of their proposed algorithm, as well as establish worst-case lower bounds, both for value and policy learning.

**Audience:**

Yes

**Audience Explanation:**

This paper would be of interest for individuals interested in RL and risk-aware decision-making.

**Claims And Evidence:**

No

**Claims Explanation:**

The authors make three claims in this paper:

1) That their proposed Model-Based ERM Q-Value Iteration algorithm is novel.

2) That they establish appropriate PAC-style bounds on the proposed algorithm.

3) That they establish appropriate lower bounds on the proposed algorithm.

In terms of the first claim, the authors claim that their proposed MB-ERM-QVI is novel in comparison to prior work (e.g. [1, 2]). However, the authors do not clearly specify the specific differences between their algorithm and the aforementioned prior work. For example, it appears that in comparison to [1], their algorithm looks at state-action value iteration rather than state-value iteration. It is also not clear how their algorithm compares to the generic framework discussed in [2]. Moreover, in Section 2, where the authors do discuss prior work, they mention that prior work only proposed planning algorithms but not learning algorithms. Yet, a value iteration algorithm (which the authors propose) is commonly viewed as a planning method since it requires a model of the environment. Overall, given this confusion, I do not find that the first claim is adequately supported.

In terms of the second and third claims, the authors derive the claimed PAC and lower bounds for value and policy learning in Theorems 1 through 4, with additional proofs in the appendix. In my view, the authors adequately show the claimed contributions related to the bounds via these theorems.

[1] Jia Lin Hau, Marek Petrik, and Mohammad Ghavamzadeh. “Entropic risk optimization in discounted MDPs”. International Conference on Artificial Intelligence and Statistics. 2023

[2] Nicole Bäuerle and Alexander Glauner. “Markov decision processes with recursive risk measures”. European Journal of Operational Research. 2022.

**Requested Changes:**

From a writing perspective, the paper is very well-written and I have no concerns in this regard. The figure included in the text is also clear and easy to understand.

My concerns/requests/questions are as follows:
- To properly support the novelty of the proposed algorithm, the authors need to be very clear and explicit about how their algorithm differs from prior work. Moreover, do these prior algorithms have any bounds associated with them that the authors could compare to?
- Typo at the start of page 2: “may lead to time-consistent optimal policies” -> should be time-inconsistent.
- In Table 1, it is not clear whether the bounds for the risk-neutral algorithms are for value and/or policy learning.
- In page 6, why is it necessary to have both the Bellman optimality equation, $Q*$, and the Bellman equation under a given policy, $Q^\pi$. Could one not simply use the optimality equation directly? If so, what is the purpose of including $Q^\pi$?
- The use of the term “Episodic MDP” can be a bit ambiguous, an alternative phrasing to use could be “Total-Reward” or "Undiscounted" MDPs.
- It is not clear how Algorithms 1 and 2 are related. For instance, should not the input to Algorithm 2 read $\hat{\mathcal{M}}$ instead of $\mathcal{M}$?
- It seems a bit underwhelming to not have any sort of empirical evaluation to validate the theoretical results. I would encourage the authors to include even a single toy experiment to strengthen their paper.

---

> ### Author Response · Authors · 2026-04-17
> **Answer to Reviewer nMh1**
>
> We thank the reviewer for the careful reading and constructive suggestion.
>
> **Comparison with [1] and [2].** We thank the reviewer for this comment.
> To the best of our knowledge, there is no prior work studying sample complexity for value or policy learning under recursive ERM in discounted MDPs. Existing work in this setting primarily focuses on planning with a known model (e.g., [2]). In contrast, our work studies the learning problem in the generative model setting, where the transition dynamics must be estimated from samples.
> Also, we note that non-recursive ERM formulations (e.g., Hau et al. [1]) optimize a different objective and are therefore not directly comparable to the recursive ERM setting considered here.
>
> Our MB-ERM-QVI algorithm is a natural extension of the value iteration scheme in [2] to a state-action (Q-function) formulation suitable for learning. Specifically, [2] considers a risk-averse setting ($\beta > 0$) in a planning context, whereas we develop a Q-value iteration scheme that applies to both risk-averse and risk-seeking regimes within a unified framework.
>
> Importantly, while the update structure is motivated by [2], its extension to a Q-function-based learning algorithm under recursive ERM in the generative setting has not been analyzed in prior work, to the best of our knowledge. In particular, we provide finite-sample PAC-type guarantees for both value and policy learning, which are not available in prior studies of this setting.
> We have revised the contribution section with more accurate pointers and statements.
>
> **About Type.** Fixed. Thanks for spotting that.
>
> **About Table 1.** We thank the reviewer for this feedback. The bounds in the risk-neutral case are similar for both value and policy learning, as it would turn out in the discussion in Related Works. We agree that the way we reported them may cause some confusion. To make things more explicit and avoid any ambiguity, we revised the table and distinguished between policy learning and value learning for the risk-neutral case.
>
> **About Bellman Equations for $Q^\pi$ and $V^\pi$.**
>
> Indeed, our analysis only relies on the Bellman optimality equation, as the reviewer has pointed out. The Bellman equation for $V^\pi$ and $Q^\pi$ is included for completeness, to help the reader contrast ERM with the risk-neutral case. We have added a numerical experiment section for which the Bellman operator for $V^\pi$ renders necessary.
>
> To avoid a potential confusion, we have highlighted this on page 6 (footnote).
>
> **About MDP Terminology.** Episodic MDP is used to refer specifically to episodic MDPs with fixed (and known) horizons, which is a widely accepted terminology in the theoretical RL community. Although these problems are formulated using the total-reward criterion, they exclude problems with random horizons (e.g., stochastic shortest path problems). Therefore, there are a subset of total-reward problems. Also we believe that “undiscounted MDPs” include average-reward MDPs problems too, which may cause further ambiguity.
>
> **Algorithm 1 and Algorithm 2 Relations.** It is correct that we use the output of Algorithm 1 as input for Algorithm 2, which is what we described in the two point steps at the bottom of page 7. In doing so, the notation $M$ was overloaded to refer to any generic MDP, yielding a generic description of Algorithm 2. We agree that this notation overload may cause confusion. In the revised version, we have made it explicit in Algorithm 2 that the input is $\widehat M$.
>
> **Addition of Numerical Experiments.** We thank the reviewer for this suggestion. In the revised paper, we have added a numerical example to illustrate the behavior of MB-ERM-QVI. Specifically, we consider the RiverSwim MDP, a standard benchmark in RL, and report policy error curves under both risk-sensitive (ERM-based) and risk-neutral objectives. The results indicate that the empirical sample complexity in the risk-sensitive setting is substantially higher than in the risk-neutral case, which aligns with the trends suggested by our theoretical results.

---

> ### Comment · Reviewer_nMh1 · 2026-04-17
>
> I thank the authors for their response to my comments as well as for the updates to the paper.
>
> Overall, the authors' response has addressed my concerns, with the exception of the comparison to [2]. In particular, I was hoping the authors could further elaborate on this point: "Importantly, while the update structure is motivated by [2], its extension to a Q-function-based learning algorithm under recursive ERM in the generative setting has not been analyzed in prior work":
> - Has an equivalent analysis been performed when the model is known beforehand? If so, how does it compare to the analysis performed in this paper?
> - To what extent does the analysis performed in this paper consider the generative aspect of the model? In particular, the authors argue that the generative aspect of their method is sufficient to distinguish it from the framework proposed in [2]. If the generative aspect meaningfully affects the analysis, then I would be inclined to agree with the authors. However, if the analysis does not meaningfully consider the generative aspect, then I would argue that the MB-ERM-QVI algorithm is just an instance of the framework proposed in [2], and that this work can be interpreted as a (high quality) derivation of bounds pertaining to the framework proposed in [2].
>
> I again thank the authors and look forward to future discussions.

---

> > ### Author Response · Authors · 2026-04-21
> > **Response to Reviewer nMh1**
> >
> > We thank the reviewer for their feedback on our responses and the follow-up question.
> >
> > First, we highlight that the main point of [2] is to show that when the model is known and the criterion is recursive with respect to a class of risk measures, there exists an optimal policy and optimal value function that satisfies a Bellman equation. This is done in the risk-averse setting that includes ERM with $\beta>0$. We need two extensions that are not covered in [2]: we need this to hold for ERM for $\beta<0$, and we need to extend the results to hold for state-action value functions (Q-functions) and not just V-functions.
> >
> > With that said, returning to the raised questions:
> > - No. [2] does not report any equivalent sample complexity analysis (under policy or value learning criterion). In fact, the sample complexity definitions capture the statistical hardness of the problem; under the known model (e.g., in [2]), there is no statistical hardness.
> > - In our sample complexity analysis, under value learning, we have to control two terms -- see Equation 8 --
> > $||\widehat{Q}^{\pi*}-Q^*||$ and  $||Q_k - \widehat{Q}^\star||$ and
> > The first term captures the statistical hardness and arises because of the generative model. We provide a careful analysis showing how many samples are needed to guarantee that this term is sufficiently small with high probability. For this part we exploit that we have a generative model available. The second term is purely computational and can be made small enough by doing value-iteration enough times, and its analysis follows from the contraction property of ERM, which is also present in the case of known model. We note that the first term is the primary component of the sample complexity bound.
> >
> > Similarly, in the case of policy learning, as shown in Equation (9), we need to control $||V^{\pi_k} - \widehat{V}^{\pi_k}||$, $||\widehat{V}^{\pi*}-V^* ||$, and $||\widehat{V}^{\pi_k} - \widehat{V}^* ||$ where the first two terms arise due to the generative model (statistical terms), and are primary component in the resulting bound.
> >
> > We would be happy to provide further clarification on any points that may be unclear.

---

> > > ### Comment · Reviewer_nMh1 · 2026-04-21
> > >
> > > I thank the authors for their clarifications.
> > >
> > > Overall, given this new explanation by the authors, all my concerns have been addressed.
> > >
> > > I strongly encourage the authors to incorporate these clarifications into the main text.

---

> > > > ### Author Response · Authors · 2026-04-28
> > > > **Revised Paper**
> > > >
> > > > We would like to thank the reviewer for further feedback.
> > > >
> > > > Following the reviewer's suggestion, we have revised the paper and included the clarifications from the last discussion in the main text (in Related Work as well as in Section 6, below Equations (8) and (9)).

---

### Review · Reviewer_T3it · 2026-03-04

**Summary Of Contributions:**

This paper studies risk-sensitive reinforcement learning in discounted tabular MDPs using recursive entropic risk measure (ERM).
The authors propose a simple model-based algorithm (MB-ERM-QVI) in the generative model setting and provide PAC upper bounds for both value learning and policy learning. They then provide Matching lower bounds. They provide a proof that exponential dependence on β / (1-γ) is unavoidable, And the recursive ERM makes learning harder than the standard risk-neutral case.

The motivation is clear, the authors provide good theoretical contributions including upper and lower bounds. They claim the sample complexity grow exponentially. They also consider and study both risk averse and risk seeking perspectives.

However the algorithm seem very simple, it basically:
1. Estimate transition probabilities uniformly
2. Run Q-value iteration
 The bound contains term e^(2∣β∣/(1−γ)) which grows extremely fast. Even though the lower bound shows exponential growth is unavoidable, there is still a noticeable gap between upper and lower bounds. More discussion is needed on whether the bounds can be tightened.

The lower bound also becomes vacuous when ∣β∣ is small. And It is unclear whether the problem actually becomes easier for small β? Or the proof technique simply breaks?

The results are provided just in the generative model setting, which is the easiest setting in RL. There is no discussion of any other setting which in my opinion limits practical relevance. More clarification in my pinion is needed about how these claims can be extended to other settings in RL

However I’m not an expert in the field, I would appreciate more explanation and Clarification on my concerns mentioned above.

**Audience:**

Yes

**Audience Explanation:**

i'm not expert in this filed but i believe talking about practical relevance is very limited in this paper

**Claims And Evidence:**

Yes

**Claims Explanation:**

i'm not a mathematician in general and also there is not much evidential results shown to be considered as a proof of concept here. However i guess they should be correct.

**Requested Changes:**

See above

---

> ### Author Response · Authors · 2026-04-17
> **Answer to Reviewer T3it**
>
> We thank the reviewer for the careful reading and constructive suggestion.
>
> **About Discussion on Tightening the Bounds.** We agree that there remains a gap between the upper and lower bounds. A useful sanity check is the risk-neutral limit as $\beta \to 0$, where the problem reduces to the standard setting. In this regime, our upper bound does not fully match the known optimal risk-neutral sample complexity, which suggests that the analysis is not tight in all parameters. In particular, this mismatch indicates that some of the dependence in the upper bound is likely an artefact of the current proof technique rather than an intrinsic difficulty of the problem. We therefore believe that there is room for improvement in the upper bound analysis, although identifying the precise source of slack remains an open question.
>
> In the risk-neutral setting, minimax-optimal sample complexity results rely on techniques that crucially exploit the additive structure of the return (see Remark 3). In contrast, in the ERM setting considered here, the objective is highly non-linear, and analogous variance reduction techniques are not directly applicable. At present, it remains unclear how to extend these tools to this setting, or whether alternative techniques could recover the minimax-optimal dependence.
>
> We view tightening the gap between upper and lower bounds as an interesting direction for future work.
>
> **About the Lower Bound in the Small $\beta$ Regime.** It is correct that the stated lower bound becomes vacuous for $|\beta| < (1-\gamma)\log(3)$, as also discussed in Remark 5. This behavior is primarily due to the specific hard instance construction used in the proof. In particular, the construction relies on a Bernoulli transition with parameter $p$, and the resulting lower bound scales with $p(1-p)$. As $|\beta|$ becomes small, the corresponding choice of $p$ approaches either $0$ or $1$, causing this term to vanish and leading to a degenerate bound. This is therefore a limitation of the current construction rather than evidence of a fundamental simplification of the problem.
>
> Whether a sharper construction could recover a non-trivial lower bound in the small-$|\beta|$ regime (and in particular interpolate more smoothly to the risk-neutral case as $\beta \to 0$) is an interesting open question.
>
> We have clarified this point in Section 9 of the revised manuscript.
>
> **About Extensions and Implications beyond the Generative Setting.** The generative model setting is commonly used in theoretical RL as it provides a clean framework in which the statistical difficulty of the problem can be isolated and precisely characterized. In particular, it removes effects due to exploration and data collection, allowing us to focus on the intrinsic difficulty induced by the risk-sensitive objective. While our results are derived in this setting, they capture fundamental estimation and distinguishability phenomena that are also present in more general RL settings. In this sense, the lower bounds should be interpreted as characterizing intrinsic statistical limitations rather than artifacts of the generative model.
>
> Regarding extensions, a natural next step is offline RL, where, under a model-based approach, similar plug-in estimators combined with pessimism-style corrections are often used. We expect that analogous challenges related to accurately estimating transition dynamics under a non-linear objective would also arise in that setting, although a precise characterization is beyond the scope of this work.
>
> In addition, some of the technical tools developed in this paper (e.g., the simulation lemma for ERM-based value estimation) may be of independent interest and could potentially be useful in analyzing related offline RL settings, although establishing such connections would require additional work.
>
> We view extending these results to offline and online RL settings as an interesting direction for future research.
> We have included some of these discussions in Section 9 of the revised manuscript.

---

### Review · Reviewer_EBUa · 2026-04-03

**Summary Of Contributions:**

The paper addresses risk-sensitive reinforcement learning in finite discounted MDPs where the objective is defined by a recursive entropic risk measure (ERM) parameterized by $\beta$ (risk-aversion for $\beta>0$, risk-seeking for $\beta<0$, risk-neutral for $|\beta|\rightarrow0$) and with access to a generative model (a generative model hypothesis). The authors propose a model-based ERM $Q$-value iteration algorithm (MB-ERM-QVI) that uses an empirically estimated MDP and applies a value iteration algorithm for $Q$-value function under ERM. The authors derive PAC-type sample complexity bounds on $T = NSA$ model calls for estimating the $\varepsilon$-optimal $Q$-value function and for obtaining an $\varepsilon$-optimal $V$ function under ERM. The upper bounds scale approximately
$$\widetilde{\mathcal{O}}\left(\frac{SA}{\varepsilon^2(1-\gamma)^2|\beta|^2}e^{2|\beta|/(1-\gamma)}\right)$$
for $Q$-value function learning and
$$\widetilde{\mathcal{O}}\left(\frac{SA}{\varepsilon^2(1-\gamma)^2|\beta|^2}\min\left\\{S,\frac{1}{(1 - \gamma)^2}\right\\}e^{2|\beta|/(1-\gamma)}\right)$$
for $V$ function learning, while matching lower bounds of order
$$\widetilde\Omega\left(\frac{SA}{\varepsilon^2(1-\gamma)^2|\beta|^2}e^{|\beta|/(1-\gamma)}\right).$$
These upper bounds are proved for $\beta\in\mathbb{R}\setminus\{0\}$ and lower bounds are proved for $|\beta| > (1 - \gamma)\ln(3)$,  both showing exponential dependence on the effective horizon $|\beta|/(1-\gamma)$. These results are claimed the first sample complexity guarantees for recursive ERM in both risk-averse and risk-seeking cases.

**Audience:**

Yes

**Audience Explanation:**

Yes, the paper has rigorous sample‑complexity results for risk‑sensitive reinforcement learning within generative model setting using the recursive entropic risk measure, including a model‑based $Q$-value iteration algorithm under this setting, and provides matching upper and lower bounds on the number of samples needed to learn value functions, therefore obtaining $\varepsilon$-optimal policies. Such contributions are directly relevant to reinforcement‑learning theory in risk‑aware decision making.

**Broader Impact Concerns:**

No broader impact concern has been found

**Claims And Evidence:**

Yes

**Claims Explanation:**

The claims made in the paper appear to be supported by sufficiently convincing and accurate evidence. The authors provide formal definitions of the entropic risk measure and derive Bellman optimality equations for discounted Markov decision processes under this risk criterion, establishing the contraction properties of the corresponding dynamic programming operators. They propose a model-based algorithm, MB-ERM-QVI, prove PAC-type sample complexity bounds for both $Q$-value learning and $V$ function estimation. These upper bounds are derived via standard MDP error decompositions, a smoothness analysis of the entropic risk operator, and concentration inequalities. The paper also has a hard MDP instance to establish minimax lower bounds on the sample complexity, yet with restriction to $|\beta| > (1 - \gamma)\ln(3)$. The hard instance demonstrates that the exponential dependence on the effective horizon $|\beta|/(1-\gamma)$ is unavoidable. The mathematical arguments are rigorous enough and internally consistent. The key claims regarding sample complexity and the necessity of exponential factors are clearly stated and proved. The nuances on $\beta$ dependency are explicitly acknowledged in remarks.

**Requested Changes:**

1) While the paper is purely theoretical an (numerical) example on how MB-ERM-QVI procedure works on hard instance would help to understand the papers' main contributions better.

2) Provide a more detailed discussion on how the presented algorithms 1 and 2 behave in the case $|\beta|\leq (1 - \gamma)\ln(3)$. Some examples on how these algorithms learn MDP might give a hint how to estimate lower bound in such case. The paper would only benefit from this.

3) In the limit case $|\beta|\rightarrow0$ the paper shows a recovery of risk neutral PAC-type bound on sample complexity which is off the minimax-optimal by a factor of $(1 - \gamma)^{-1}$. Possibly a refined analysis (eg. using variance reduction techniques similarly to risk-neutral RL) would help to either match the existing minimax-optimal estimates in the limit or show us why this factor is present.

---

> ### Author Response · Authors · 2026-04-17
> **Answer to Reviewer EBUa**
>
> We thank the reviewer for the careful reading and constructive suggestion.
>
> **About Item 1.** In the revised paper, we have added a numerical example to illustrate the behavior of MB-ERM-QVI (Section 8). Specifically, we consider the RiverSwim MDP, a standard benchmark in RL, and report policy error curves under both risk-sensitive (ERM-based) and risk-neutral objectives. The results indicate that the empirical sample complexity in the risk-sensitive setting is substantially higher than in the risk-neutral case, which is qualitatively consistent with our analysis.
>
> Regarding the suggestion to include experiments on the hard instance, we would like to clarify a subtle but important issue. In our setting, the construction of a hard instance MDP depends explicitly on the parameters $\beta$ and $\gamma$. As a result, varying these parameters (e.g., to study their effect on performance) would necessarily lead to different underlying MDPs. This makes it difficult to carry out meaningful empirical comparisons, as performance differences could stem from changes in the MDP itself rather than from the algorithmic behavior. For this reason, we believe that experiments on standard benchmark environments such as RiverSwim provide a more interpretable and controlled way to illustrate the practical implications of our theoretical results.
>
> **About Item 2.** Our algorithm is model-based, so it first constructs an empirical estimate of the MDP (i.e., transition probabilities and rewards) and then computes a policy by applying a QVI procedure to this estimated model. Its performance therefore depends critically on how accurately the underlying MDP can be learned from finite samples.
> To provide intuition in the context of the hard instance, the key difficulty lies in estimating transition probabilities that are deliberately chosen to be very close (e.g., distinguishing between $p$ and $p+\alpha$ for small $\alpha$). When the number of samples is below a certain threshold, the algorithm cannot reliably distinguish between two such instances, say $M_0$ and $M_1$. As a result, the empirical model constructed by the algorithm may be inaccurate in a way that leads to selecting suboptimal policies.
>
> To clarify this point, we have added a short paragraph – after the proof sketch and before the two remarks – that discusses algorithmic intuition for how a model-based procedure behaves on the hard instance and how the indistinguishability argument manifests in terms of transition estimation.
>
> **About Item 3.** We agree that the extra factor in the limit case is likely an artefact of the proof technique rather than a fundamental limitation. In the risk-neutral setting, minimax-optimal sample complexity results rely on techniques that crucially exploit the additive structure of the return (e.g., variance-based arguments and its Bellman consistency properties). In contrast, in the ERM setting considered here, the objective is highly non-linear, and analogous variance reduction techniques are not directly applicable. At present, it remains unclear how to extend these tools to this setting, or whether alternative techniques could recover the minimax-optimal dependence.
>
> This point is already discussed in Remark 3. We believe that developing sharper analyses in this direction is an interesting direction for future work.

---

### Decision · Action_Editor_dDY8 · 2026-05-18

**Recommendation:** Accept as is

**Audience:**

Yes

**Audience Explanation:**

The reviewers agree that the paper is of interest for the community.

**Claims And Evidence:**

Yes

**Claims Explanation:**

After the interaction with the authors, the reviewers think that the paper makes claims that are convincing and supporeted by evidence.